# Histone H3 clipping is a novel signature of human neutrophil extracellular traps

Dorothea Ogmore Tilley[1], Ulrike Abuabed[2], Ursula Zimny Arndt[3], Monika Schmid[3], Stefan Florian[4], Peter R Jungblut[3], Volker Brinkmann[2], Alf Herzig[1†], Arturo Zychlinsky[1*†]

[1]Department of Cellular Microbiology, Max Planck Institute for Infection Biology, Berlin, Germany; [2]Microscopy Core Facility, Max Planck Institute for Infection Biology, Berlin, Germany; [3]Protein Analysis Core Facility, Max Planck Institute for Infection Biology, Berlin, Germany; [4]Institut für Pathologie, Charité - Universitätsmedizin Berlin, Berlin, Germany

**Abstract** Neutrophils are critical to host defence, executing diverse strategies to perform their antimicrobial and regulatory functions. One tactic is the production of neutrophil extracellular traps (NETs). In response to certain stimuli, neutrophils decondense their lobulated nucleus and release chromatin into the extracellular space through a process called NETosis. However, NETosis, and the subsequent degradation of NETs, can become dysregulated. NETs are proposed to play a role in infectious as well as many non-infection related diseases including cancer, thrombosis, autoimmunity and neurological disease. Consequently, there is a need to develop specific tools for the study of these structures in disease contexts. In this study, we identified a NET-specific histone H3 cleavage event and harnessed this to develop a cleavage site-specific antibody for the detection of human NETs. By microscopy, this antibody distinguishes NETs from chromatin in purified and mixed cell samples. It also detects NETs in tissue sections. We propose this antibody as a new tool to detect and quantify NETs.

*For correspondence:
zychlinsky@mpiib-berlin.mpg.de

†These authors contributed equally to this work

## Editor's evaluation

This study presents and characterizes a new antibody to detect human neutrophil extracellular traps (NETs). The authors identify a NET-specific histone H3 cleavage and make use of this event to develop a cleavage site-specific antibody. This new tool should be useful to investigators interested in detecting and quantifying human NETs.

## Introduction

Neutrophil extracellular traps (NETs) are extracellular structures consisting of chromatin components, including DNA and histones, and neutrophil proteins (*Brinkmann et al., 2004*; *Urban et al., 2009*). NETs were first described as an antimicrobial response to infection, facilitating trapping and killing of microbes (*Brinkmann et al., 2004*). They are found in diverse human tissues and secretions where inflammation is evident (recently reviewed by *Sollberger et al., 2018*). NETs are produced in response to a wide-range of stimuli; bacteria *Brinkmann et al., 2004*; fungi *Urban et al., 2006*; viruses *Saitoh et al., 2012*; *Schönrich et al., 2015*; crystals *Schauer et al., 2014*; and mitogens (*Amulic et al., 2017*). Both NADPH oxidase (NOX)-dependent and NOX-independent mechanisms lead to NET formation (*Bianchi et al., 2009*; *Hakkim et al., 2011*; *Kenny et al., 2017*; *Neeli and Radic, 2013*). NETs are also observed in sterile disease, including multiple types of thrombotic disease (recently reviewed by *Jimenez-Alcazar et al., 2017*) and even neurological disease (*Zenaro et al., 2015*). NETs, or their

components, are implicated in the development and exacerbation of autoimmune diseases including psoriasis, vasculitis, and systemic lupus erythematosus (recently reviewed by *Papayannopoulos, 2018*) as well as cancer and cancer metastasis (*Albrengues et al., 2018*; *Cools-Lartigue et al., 2013*; *Demers et al., 2016*). Consequently, there is an urgency across multiple fields to establish the pathological contribution of NETs to disease. However, the detection of NETs in affected tissues remains a challenge.

NETs are histologically defined as areas of decondensed DNA and histones that colocalise with neutrophil granular or cytoplasmic proteins. Thus, reliable detection of NETs requires a combination of anti-neutrophil and anti-chromatin antibodies as well as DNA stains. Immunofluorescent microscopy is a useful method to detect NETs in tissue sections and in vitro experiments. However, this can be challenging since NET components are distributed across the large decondensed structure resulting in a weak signal. For example, the signal of antibodies to neutrophil elastase (NE) is significantly dimmer in NETs than in the granules of resting cells where this protease is highly concentrated. Conversely, anti-histone antibodies stain NETs strongly but not nuclei of naïve neutrophils, where the chromatin is compact and less accessible. This differential histone staining property can be exploited for the detection and quantification of NETs (*Brinkmann et al., 2012*). However, sample preparation and the subsequent image analysis make results between different labs difficult to compare. Thus, there is a need to identify antibodies against NET antigens.

NETs can also be detected through post-translational modifications (PTMs) that occur during NETosis. Histone 3 (H3) is deaminated in arginine residues - the conversion to citrulline (citrullination) - by protein arginine deiminase 4 (PAD4) (*Wang et al., 2009*). Citrullinated H3 (H3cit) is widely used as a surrogate marker of NETs in both in vitro and in vivo experiments (*Gavillet et al., 2015*; *Pertiwi et al., 2018*; *Wang et al., 2009*; *Yoo et al., 2014*; *Yoshida et al., 2013*). Cleavage of histones by granular derived neutrophil serine proteases (NSPs) also contributes to NETosis (*Papayannopoulos et al., 2010*). Histone cleavage, or clipping, by cysteine or serine proteases is a bona fide histone PTM that facilitates the gross removal of multiple, subtler, PTMs in the histone tail and is conserved from yeast to mammals (*Dhaenens et al., 2015*). Until now, histone clipping has not been exploited for the detection of NETs but recent work by our group showed that histone H3 cleavage is a conserved response to diverse NET stimuli, including *Candida albicans* and Group B Streptococcus (*Kenny et al., 2017*). Thus, in this study we map the site(s) of histone H3 cleavage during NETosis. We developed a new monoclonal antibody against cleaved H3 that detects human NETs in vitro and in histological samples. This antibody also facilitates easier NET quantification.

## Results
### Serine protease dependent cleavage of Histone H3 N-terminal tails during NET formation

Histones are processed in response to phorbol 12 myristate 13 acetate - PMA (*Papayannopoulos et al., 2010*; *Urban et al., 2009*) and other NET stimuli (*Kenny et al., 2017*). Indeed, the cleavage products of H3 were consistent between stimuli (*Kenny et al., 2017*). This suggests that H3 proteolysis occurs at specific sites during NETosis. In a time course experiment of human neutrophils incubated with PMA, we detected a H3 cleavage product of ~14 kDa as early as 30 min post-stimulation (*Figure 1A*). Further cleavages occurred between 60 and 90 min, yielding products of approximately 13 kDa and 10 kDa, respectively. The histone N-terminal tails protrude from the nucleosome core and are a major PTM target (*Bannister and Kouzarides, 2011*). A C-terminal, but not an N-terminal, directed histone antibody detected the cleavage products of H3 (*Figure 1A*). These results indicate that the N-terminus is cleaved in truncated H3.

Neutrophil azurophilic granules are rich in serine proteases that can degrade histones in vitro (*Papayannopoulos et al., 2010*). We tested the contribution of these proteases to H3 cleavage during NET formation. Preincubation with the serine protease inhibitor, AEBSF (4- [2-aminoethyl] benzensulfonylfluoride) (*Figure 1A*), but not with the cysteine protease inhibitor E64 (*Figure 1—figure supplement 1*), inhibited H3 cleavage upon PMA stimulation. AEBSF also inhibited NET formation and nuclear decondensation as shown by immunofluorescent microscopy (*Figure 1B*). PMA induced NETosis requires NADPH oxidase activity and, at high concentrations, AEBSF can inhibit activation of NAPDH oxidase (*Diatchuk et al., 1997*). To rule out this upstream effect, we showed that AEBSF did

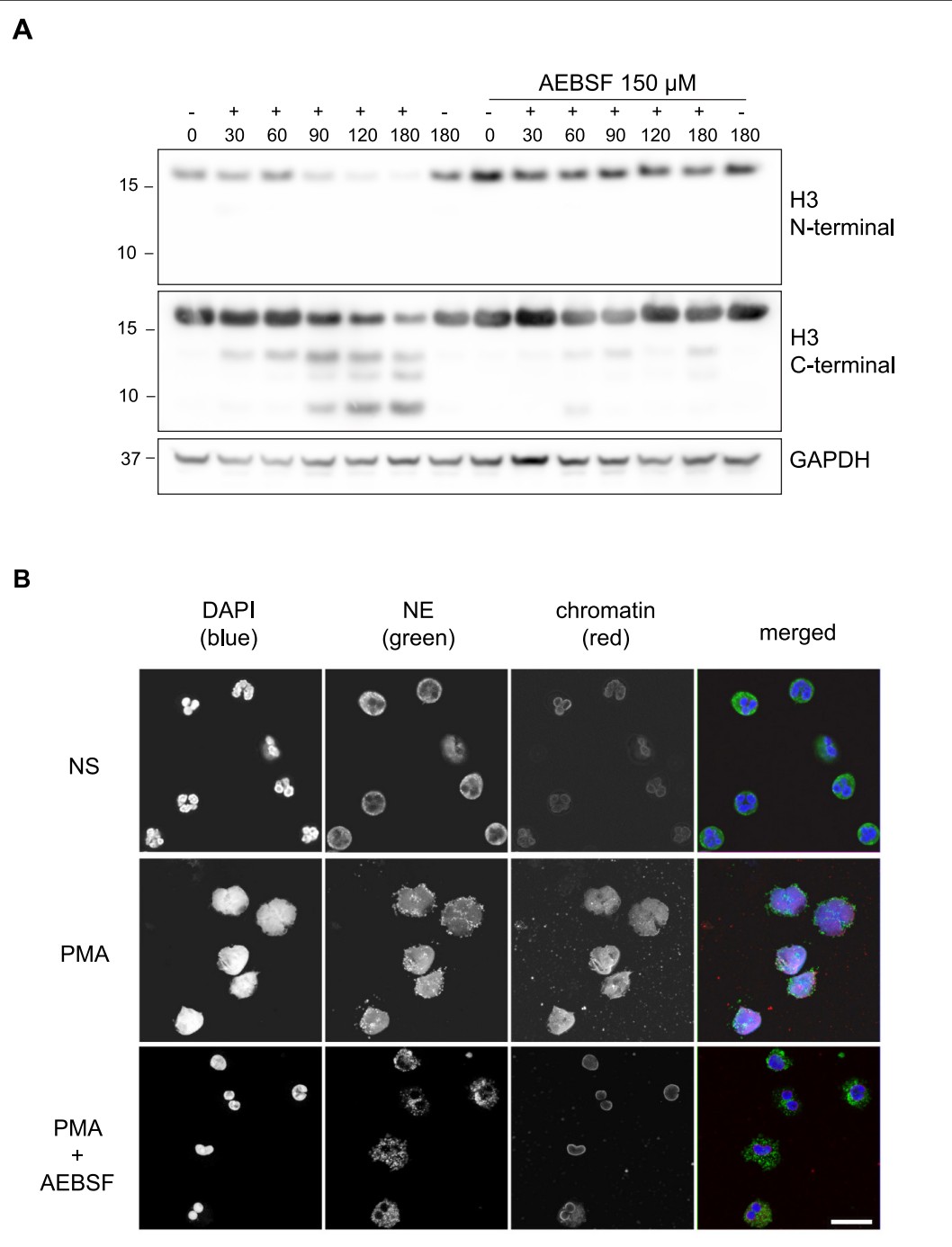

**Figure 1.** PMA-induced Histone H3 cleavage occurs in the N-terminal domain and is prevented by serine protease inhibition. (**A**) Neutrophils were preincubated with the serine protease inhibitor AEBSF for 30 min in microcentrifuge tubes and then stimulated with PMA as indicated in the figure. Lysates were resolved by SDS-PAGE and immunoblotted with N- and C-terminal antibodies to H3. GAPDH was used as a loading control. Blots representative of three independent experiments. (**B**) Immunofluorescent confocal microscopy of NET formation. Neutrophils were seeded on coverslips and preincubated with AEBSF before stimulation with PMA (50 nM) as indicated in the figure. At 150 min the cells were fixed and stained for neutrophil elastase (NE), chromatin (using a H2A-H2B-DNA antibody PL2.3) and DNA (DAPI [4',6-diamidino-2-phenylindole]). NS: non-stimulated. Images were taken using at 63 x (Plan Apochromat, glycerol, numerical aperture 1.30) and the scale bar is 20 µm. Images are representative of three independent experiments. Source data is found in *Figure 1—source data 1*.

The online version of this article includes the following source data and figure supplement(s) for figure 1:

*Figure 1 continued on next page*

*Figure 1 continued*

**Source data 1.** Time course of H3 cleavage and inhibition by AEBSF.

**Figure supplement 1.** Cysteine protease inhibition by E64 does not inhibit histone H3 cleavage.

**Figure supplement 1—source data 1.** Cysteine protease inhibition does not inhibit Histone H3 cleavage.

**Figure supplement 2.** AEBSF does not inhibit ROS production.

**Figure supplement 2—source data 1.** AEBSF does not inhibit the ROS burst.

**Figure supplement 3.** AEBSF is not cytotoxic.

**Figure supplement 3—source data 1.** AEBSF is not cytotoxic.

not inhibit ROS production at the concentrations used in our assay (*Figure 1—figure supplement 2*). Similarly, we verified that at these concentrations AEBSF was not cytotoxic, as shown by limited LDH release (*Figure 1—figure supplement 3*). This data shows that the N-terminus of H3 is cleaved early during NET formation and that this event is dependent on serine protease activity.

## Histone H3 is cleaved at a novel site in the globular domain

To identify the precise H3 cleavage sites we prepared histone enriched extracts from primary neutrophils stimulated with PMA for 90 min and then purified H3 by RP-HPLC as previously described (*Shechter et al., 2007*). A schematic summary of this is presented in *Figure 2—figure supplement 1*. We identified the fractions containing H3 and its cleaved products by Western blot with anti-H3 C-terminal antibodies (*Figure 2A*). As expected, H3 was the last core histone to elute (at 45–46 min).

We further separated H3 and its truncated forms by two-dimensional electrophoresis (2-DE) and confirmed their identity by mass spectrometry (*Figure 2B* and *Figure 2—figure supplement 2* and *Table 1*). The sequence coverage did not include residues that allowed the differentiation of H3 variants. The N-terminals of the separated H3 fragments were not covered by MS and therefore sequenced by Edman degradation from two independent experiments (*Table 2*). The N-terminal sequence of the largest molecular weight H3 spot matched that of intact H3 (*Figure 2B*, spot 1). We did not obtain reliable sequencing of spots 2 and 3 but the cleavage sites of spots 4 and 5 were identified. The most truncated H3 fragment (spot 5) was cleaved between L48 and R49, in the globular domain of the protein, within the nucleosome core structure (*Figure 2C*). This is a previously unidentified cleavage site in H3 and, thus, we selected cleavage at H3R49 as a candidate marker of NETs.

## Generation of a histone H3 cleavage site monoclonal antibody

We adopted a similar strategy to *Duncan et al., 2008* to raise antibodies against the cleaved site. We designed a lysine branched immunogen containing the five amino acids at the carboxylic side of the H3R49 cleavage site (outlined in *Figure 3—figure supplement 1* and *Table 3*) and used it to immunize mice. After preliminary screening by ELISA against the immunogen and control peptides, we selected sera, and later hybridoma clones, that detected cleaved H3 in immunoblots of PMA stimulated cell lysates. We excluded sera and clones that detected full length histone H3 in addition to cleaved H3 (*Figure 3A*). We selected sera and clones that detected NETs but not resting chromatin of naïve neutrophils by immunofluorescence microscopy. Of the six mice immunised, we obtained one stable clone, 3D9, that functioned in both Western blot and microscopy – other clones performed only in Western blot (data not shown). 3D9 recognised a protein of ~10 kDa in neutrophils stimulated with PMA for 120 min and longer but did not detect any protein in resting or early stimulated cells (*Figure 3A*). This band corresponded in size with the smallest H3 fragment detected by the H3 C-terminal antibody. Interestingly, by microscopy, 3D9 exclusively recognised neutrophils undergoing NETosis – with decondensing chromatin (*Figure 3B*). To test the binding of the antibody to the de novo N-terminal H3 epitope of NETs, we performed competition experiments with the immunising peptide. In contrast to a control peptide, saturation of 3D9 with the immunising peptide blocked binding to NETs as shown by immunofluorescent microscopy (*Figure 3—figure supplement 2*).

3D9 binds specifically to cleaved H3. This antibody binds to the immunizing peptide and to isolated NETs, but not to equal concentrations of chromatin, recombinant H3 or purified calf thymus DNA, by direct ELISA (*Figure 3C*). Furthermore, in immunoprecipitation experiments, 3D9, but not an isotype control, pulled down intact H3 in lysates of naïve and activated neutrophils, but only the cleaved fragment from activated cells (*Figure 3—figure supplement 3*). We detected these pull downs both

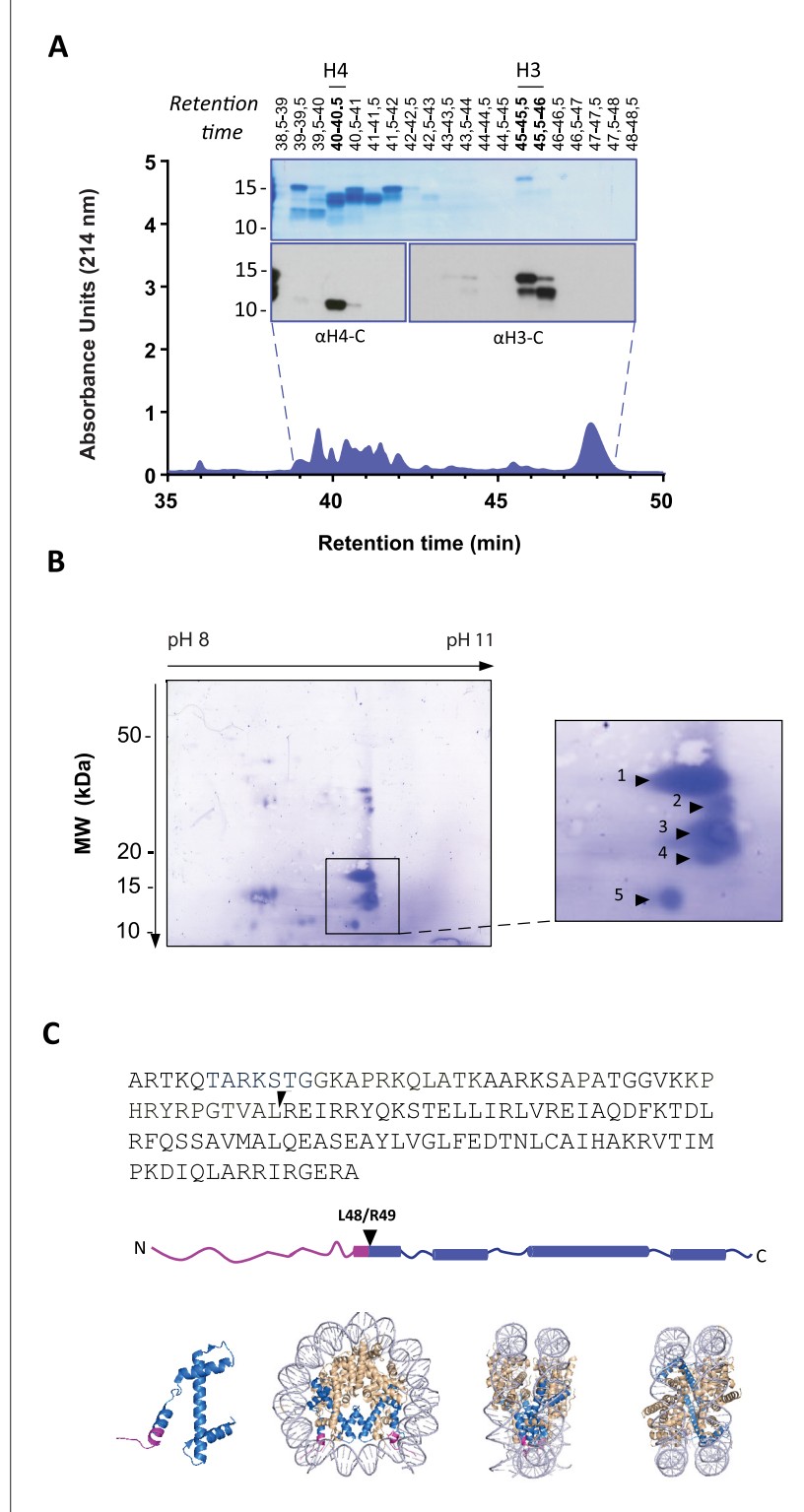

**Figure 2.** Identification of histone H3 cleavage sites in NET formation. (**A**) Representative RP-HPLC chromatogram of acid extracted histones from NETs and corresponding 1D-SDS-PAGE and immunoblots to identify H3 and H4 containing fractions. Histone enriched supernatants were prepared from neutrophils stimulated with PMA for 90 min. Purification and subsequent 2D-electrophoresis (2D-E) was repeated three times with independent donors. Source data is found in *Figure 2—source data 1* and *Figure 2—source data 2*. (**B**) Representative Coomassie stained blot of pooled H3 fractions separated by 2D-E. Inset is a zoom of all spots (1-5) identified as histone H3 by

*Figure 2 continued on next page*

*Figure 2 continued*

mass spectrometry. Other proteins identified are listed in *Figure 2—figure supplement 2* and *Table 1*. Source data is found in *Figure 2—source data 3*. (**C**) Schematic representation of the cleavage site of the truncated H3 product in both the linear sequence of H3 and in the nucleosomal context. H3 is represented in blue and the pink tail region and partial alpha helix represent the part of H3 that is removed. The nucleosome structure is adapted from PDB 2F8N (*Chakravarthy and Luger, 2005*).

The online version of this article includes the following source data and figure supplement(s) for figure 2:

**Source data 1.** Western blots and gels for histones H3 and H4 in RP-HPLC fractions.

**Source data 2.** RP-HPLC absorbance curve.

**Source data 3.** 2D-E blots for Edman degradation.

**Figure supplement 1.** Schematic summary of extraction and identification of histone H3 cleavage sites in NETs.

**Figure supplement 2.** Separation of histone H3 by two dimensional electrophoresis.

**Figure supplement 2—source data 1.** 2DE-gel showing spots for identification by mass spectrometry.

by Coomassie and silver stained gels. In a similar experiment, we immunoblotted the immunoprecipitate with the C-terminal antibody as well as 3D9 (*Figure 3—figure supplement 3* [ii]) and showed that only our monoclonal antibody recognises cleaved H3. However, when we performed preliminary experiments to see if the antibody had the potential to work in blood samples, we observed a strong reaction of the antibody with a plasma protein(s), but not with serum-protein(s) as determined by direct ELISA and western blot (*Figure 3—figure supplement 4*). Therefore, 3D9 may not be suitable for direct detection of NETs in biological fluids that may contain plasma proteins. Together this data shows that 3D9 is selective for cleaved H3 under the denaturing conditions of SDS-PAGE but may also recognize full length H3 under the more native conditions of IP.

## Epitope mapping

To determine the binding site of 3D9 in histone H3, we tested the antibody by ELISA with overlapping linear peptide arrays and helical peptide mimic arrays based on a sequence (residues 30–70; PATG GVKKPHRYRPGTVALREIRRYQKSTELLIRKLPFQRL) around the H3R49 cleavage site (*Figure 4—figure*

**Table 1.** Mass spectrometry identification of proteins co-separating with histone H3 following RP-HPLC and 2-DE.

| Spot | score | accession no. | protein name | MW | pI | sequence coverage | Number of peptides |
|---|---|---|---|---|---|---|---|
| 1 | 99 | P84243 | Histone H3.3* | 15318 | 11.27 | 25.7 | 8 |
| 2 | 133 | P84243 | Histone H3.3 | 15318 | 11.27 | 25 | 7 |
| 2 | 78 | P0C0S5 | Histone H2A.Z | 13545 | 10.58 | 14.1 | 4 |
| 3 | 140 | P84243 | Histone H3.3 | 15318 | 11.27 | 22.1 | 7 |
| 4 | 193 | P84243 | Histone H3.3 | 15318 | 11.27 | 27.2 | 8 |
| 5 | 198 | P84243 | Histone H3.3 | 15318 | 11.27 | 27.2 | 9 |
| 6 | 317 | P06702 | Protein S100-A9 | 13234 | 5.71 | 70.2 | 10 |
| 7 | 486 | P06702 | Protein S100-A9 | 13234 | 5.71 | 81.6 | 15 |
| 8 | 483 | P06702 | Protein S100-A9 | 13234 | 5.71 | 82.5 | 14 |
| 9 | 57 | P31949 | Protein S100-A11 | 11733 | 6.56 | 9.5 | 2 |
| 10 | 344 | P31949 | Protein S100-A11 | 11733 | 6.56 | 45.7 | 8 |
| 11 | 343 | P05109 | Protein S100-A8 | 10828 | 6.51 | 37.6 | 9 |
| 12 | 140 | P05109 | Protein S100-A8 | 10828 | 6.51 | 25.8 | 5 |
| 13 | 136 | P25815 | Protein S100-P | 10393 | 4.75 | 35.8 | 6 |
| 14 | 48 | P06703 | Protein S100-A6 | 10173 | 5.33 | 16.7 | 3 |

**Table 2.** N-terminal histone H3 cleavage sites identified by Edman degradation sequencing.

| Spot # | Name | Round 1 | Round 2 |
|--------|------|---------|---------|
| 1 | Intact H3 | ARTKQ | ARTKQ |
| 2 | CS1 | *NA* | T |
| 3 | CS2 | AARKS | AAAAS TKRRR |
| 4 | CS3 | TGGV | TGGVK* AAA |
| 5 | CS4 | REIRR | REIRR* AL-EI |

*NA*: not analysable.
*Confirmation of first round of identification.

supplements 1 and 2). In epitope mapping, acetylation is commonly used during peptide synthesis to neutralize the positive charge of the terminal amine groups, making the peptide more closely resemble its native conformation, as part of a larger protein. However, as we were interested in a binding site formed as a consequence of proteolysis, we included both acetylated and non-acetylated arrays of peptides to allow for potential changes in the charge of the terminal amino acid that may contribute to antibody binding. Based on overlapping peptides, the putative core epitope in the linear array was (R)EIRR. The peptides ending in REIRR were in all cases in the top 2 of each peptide mimic (*Table 4*). Interestingly, peptides extended at the N-terminus were still recognized. Moreover, acetylation at the N-terminus of the peptide ending in the REIRR sequence did not affect the binding, suggesting that a free N-terminus may not be recognized by the antibody. We further refined the epitope mapping by amino acid replacement analysis of linear peptides and helical peptide mimetics ending in REIRR (*Figure 4—figure supplement 3*). Mutations in Glu51, Ile52, and Arg54 negatively impacted the signal, indicating these residues are critical for epitope recognition. A schematic of the antibody epitope mapped onto H3 is presented in *Figure 4*.

## Automatic quantification of in vitro generated NETs by microscopy

We tested how 3D9 stained NETs in immunofluorescence in samples that were robustly permeabilized (Triton X-100, 0.5% for 10 min) to facilitate the distribution of the antibody throughout the sample. We compared its staining to a control antibody commonly used in combination with neutrophil markers to detect NETs - PL2.3, an anti-chromatin antibody directed against a H2A-H2B-DNA epitope (*Losman et al., 1992*). PL2.3, in conjunction with staining for neutrophil granule proteins, is used to facilitate detection of NETs by immunofluorescent microscopy (*Brinkmann et al., 2012*). *Figure 5A* shows that 3D9 detects decondensed chromatin almost exclusively. In contrast, PL2.3 stains NETs in addition to condensed nuclei. We compared the staining characteristics of 3D9 and PL2.3 during NET formation. We determined the nuclear area and signal intensity at the indicated time points, from multiple fields of view (*Figure 5B*). Both antibodies detect the increase in nuclear area characteristic of NETosis between 15 and 180 min after simulation. At later time points, the intensity of PL2.3 staining decreased and failed to discriminate between resting cell nuclei and NETs. In contrast, 3D9 stained nuclei undergoing NETosis with greater intensity than nuclei of non-activated cells.

Publicly available software (ImageJ) can be used to quantify in vitro NETosis. We compared 3D9 versus the anti-chromatin antibody with our previously published semi-automatic image analysis (*Brinkmann et al., 2012*) and with a modified automatic method (*Figure 5—figure supplement 1*). Both methods use automatic thresholding of the DNA channel (Hoechst) to count total cells/objects. The historical semi-automatic method exploits the differential staining by chromatin antibodies of decondensed chromatin (high signal) over compact chromatin (weak signal) to count cells in NETosis. This method uses a manual thresholding and segmentation procedure (denoted manual in the figure). This manual thresholding step is subject to observer bias. In contrast, the modified method uses automatic thresholding at both stages; total cell and NET counts. Both methods use a size exclusion particle analysis step so that only structures larger than a resting nucleus are

**Table 3.** List of immunisation, screening, and competition peptides.

| | |
|--------|------|
| Immunisation | H - REIRRK(RRIER)C - NH$_2$ *KLH conjugated* |
| Screening | (+) H - REIRRK(RRIER)C - NH$_2$<br>(-) H - AARKSK(SKRAA)C - NH$_2$* |
| Validation | H - REIRRK(RRIER) - NH$_2$ *competition peptide*<br>H - TGGVKK(KVGGT) - NH$_2$ *negative control* |

*Alternative branched peptide used for screening. The N-terminal is represented by H (H$_2$N) and the C-terminal is represented by NH$_2$ (CONH$_2$).

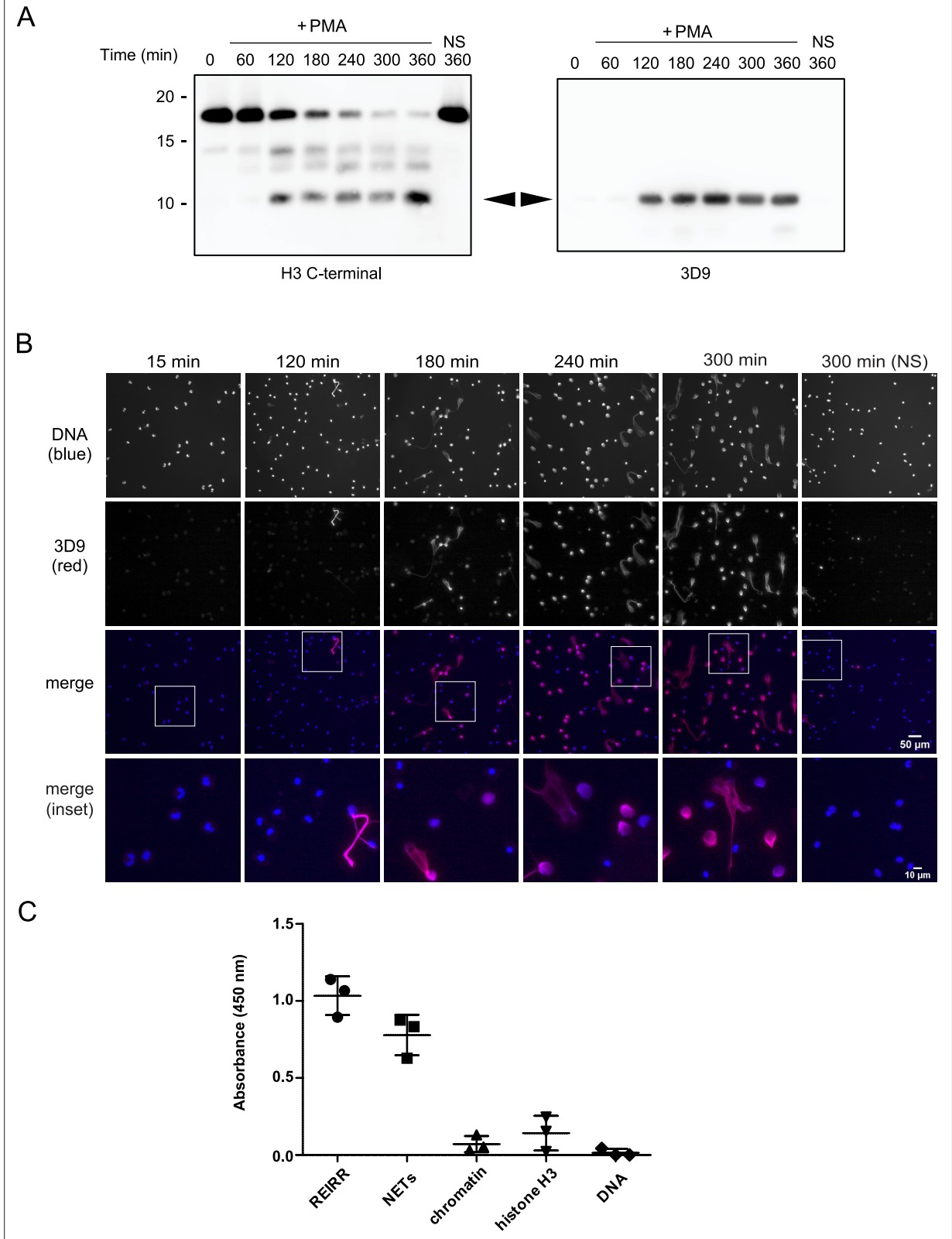

**Figure 3.** Screening and detection of cleaved H3 and NETs by 3D9. (**A**) Immunoblots of lysates prepared from neutrophils stimulated with PMA (50 nM) for the times indicated in the figure. H3 C-terminal antibody was used as a control to detect all H3 forms while a single band (cleaved H3) was detected by the newly generated monoclonal antibody, 3D9. Source data can be found in *Figure 3—source data 1*. (**B**) Immunofluorescent microscopy of neutrophils stimulated with PMA and fixed at the indicated times. Samples were stained with Hoechst (DNA - blue) and 3D9 (with Alexafluor-568

*Figure 3 continued on next page*

*Figure 3 continued*

conjugated secondary antibody - red). NS: non-stimulated. Images were taken on an upright fluorescent microscope at 20 x (Fluotar, numerical aperture 0.5). Scale bars – 50 µm (full field) and 10 µm (inset) (**C**) Direct ELISA for cleaved H3 in NETs, chromatin (A549 lung epithelial cells), recombinant histone H3 and DNA. Samples were serially diluted and immobilized on a high-affinity ELISA plate according to DNA content (for NETs, chromatin and DNA) or protein content (for recombinant histone H3) as determined by PicoGreen and bicinchoninic acid assays respectively. Starting concentration was 1 µg/ml DNA or protein. Cleaved H3 was detected using 3D9 (2 µg/ml) and HRP conjugated anti-mouse secondary antibody and reactions were developed using TMB (3,3',5,5'-tetramethylbenzidine) as a substrate. Data is presented for dilution 200 ng/ml. REIRR peptide control was coated at 20 ng/ml. Data represents mean ± SD of 3 experiments using independent NET donors. Source data can be found in *Figure 3—source data 2*.

The online version of this article includes the following source data and figure supplement(s) for figure 3:

**Source data 1.** Western blots with H3-terminal antibody and hybridoma clone 3D9.

**Source data 2.** Direct ELISA for cleaved H3.

**Figure supplement 1.** Outline of immunisation and screening strategy for antibody production.

**Figure supplement 2.** Peptide inhibition of 3D9 binding to NETs.

**Figure supplement 3.** 3D9 immunoprecipitation.

**Figure supplement 3—source data 1.** 3D9 immunoprecipitation.

**Figure supplement 4.** Detection of 3D9 cross reacting proteins from plasma and serum.

**Figure supplement 4—source data 1.** Detection of 3D9 cross-reacting proteins from plasma and serum by western blot.

**Figure supplement 4—source data 2.** Detection of 3D9 cross-reacting proteins from plasma and serum by ELISA.

counted. Both 3D9 and PL2.3 antibodies effectively quantified NETs using previously published method (*Figure 5C* - manual). However, PL2.3 failed to accurately quantify the number of NETs with the fully automatic method, specifically at early time points after stimulation (15 min). In this case, the weakly staining lobulated nucleus can extend over a larger surface area during cell activation and adhesion resulting in these cells being wrongly categorised as NETs by the algorithm. Together, this data suggests that the automatic method using 3D9 staining may reduce experimental bias.

## 3D9 detects NETs induced by multiple stimuli

Histone H3 cleavage is a feature of the neutrophil response to multiple NET stimuli (*Kenny et al., 2017*). 3D9 detects NETs induced by the bacterial toxin nigericin, which induces NETs independently of NADPH oxidase activation (*Kenny et al., 2017*), by heme in TNF primed neutrophils (*Knackstedt et al., 2019*) and by the fungal pathogen *Candida albicans* (*Figure 6*). Interestingly, in *C. albicans* infections, we observed both 3D9 positive and negative NETs. To test 3D9 with immune complex induced NETs, we attempted to induce NETs with RNP/anti-RNP (ribonucleoprotein) complexes, however there was no induction of NETs in healthy neutrophils.

## 3D9 distinguishes NETs in mixed cell samples

Histone H3 clipping, albeit at other sites in the N-terminal tail, was observed in mast cells (*Melo et al., 2014*) and unstimulated PBMC fractions (*Howe and Gamble, 2015*). Of note, PBMC fractions often contain contaminating neutrophils (*Hacbarth and Kajdacsy-Balla, 1986*). To test if 3D9 specifically stained neutrophils treated with NET stimuli, we incubated PBMCs with PMA or nigericin (*Figure 7*). Importantly, 3D9 detected only nuclei that appeared decondensed in cells that were positive for NE, a specific neutrophil marker. In contrast, the control chromatin antibody stained both neutrophils in NETosis (co-staining with anti-NE) and the nuclei of other cells (lacking NE). This shows that 3D9 detects NETs specifically even in the presence of other blood cells.

## 3D9 distinguishes NETosis from apoptotic, necroptotic, and necrotic cell death in neutrophils

Neutrophils can commit to other cell death pathways (recently reviewed by *Dąbrowska et al., 2019*) besides NETosis. Naïve neutrophils undergo apoptosis after overnight incubation (*Kobayashi et al., 2005*) and necroptosis upon TNFα stimulation in the presence of a SMAC (second mitochondria-derived activator of caspase) mimetic and if caspases are inhibited (*Galluzzi et al., 2012*). Interestingly, the control anti-chromatin antibody (*Figure 8—figure supplement 1*), but not 3D9, stained the condensed nuclei of cells undergoing apoptosis (*Figure 8*). Further staining of the apoptotic marker,

**Table 4.** Summary of identified 3D9 binding regions in the peptide array.

| Sample | Type | Peak | Epitope candidate |
|---|---|---|---|
| 3D9-1 | LIN8 | 47–58 | R**EIRR**YQK |
| | | | VALR**EIRR** |
| | | | **EIRR**YQKS |
| | | 60–68 | **LLIRKLP** |
| | | | ELLIRKLP |
| | LIN8 - Ac | 47–60 | VALR**EIRR** |
| | | | REIRRYQK |
| | | | **RR**YQKSTE |
| | | | EIRRYQKS |
| | | | LREIRRYQ |
| | | 61–68 | LLIRKLPF |
| | LIN15 | 40–64 | HRYRPGTVAL**REIRR** |
| | | | **REIRR**YQKSTELLIR |
| | | | TVAL**REIRR**YQKSTE |
| | | 1–33 | RYQKSTELLIRKLPF |
| | LIN15 - Ac | 12–44 | HRYRPGTVAL**REIRR** |
| | | | **REIRR**YQKSTELLIR |
| | | | PGTVAL**REIRR**YQKS |
| | | | VAL**REIRR**YQKSTEL |
| | | | RPGTVAL**REIRR**YQK |
| | | | YRPGTVAL**REIRR**YQ |
| | | | RYRPGTVAL**REIRR**Y |
| | | | TVAL**REIRR**YQKSTE |
| | | | AL**REIRR**YQKSTELL |

cleaved caspase 3, and 3D9 (*Figure 8—figure supplement 2*), showed that apoptotic cells did not display cleaved H3R49. Neither antibody stained cells during necrosis induced by the staphylococcal toxin α-haemolysin nor after stimulation with necroptosis inducers.

## 3D9 labels NETs in human tissue sections

Neutrophils are recruited to sites of inflammation and, depending on the context or the surrounding stimuli, they may undergo varied forms of cell death. NETs are detected in inflamed tissues based on the juxtaposition of chromatin and granular markers. Citrullination of H3 is also used in NET detection, albeit more convincingly when co-stained with a neutrophil granule or cytoplasmic marker. 3D9 stains areas of decondensed DNA (Hoechst) that colocalise with NE in both inflamed human tonsil (*Figure 9A*) and kidney (*Figure 9B*). This indicates that 3D9 can label NETs in histological samples - hematoxylin and eosin (HE) tissue overviews are provided in *Figure 9—figure supplement 1*, *Figure 10—figure supplement 1* and *Figure 11—figure supplement 1*. Indeed, in kidney (*Figure 9C*), inflamed appendix (*Figure 10*) and gallbladder (*Figure 11*), 3D9 labelled DNA in the same cluster as anti-H3cit or anti-H2B. Interestingly, 3D9 stained decondensed, more NET-like structures, while

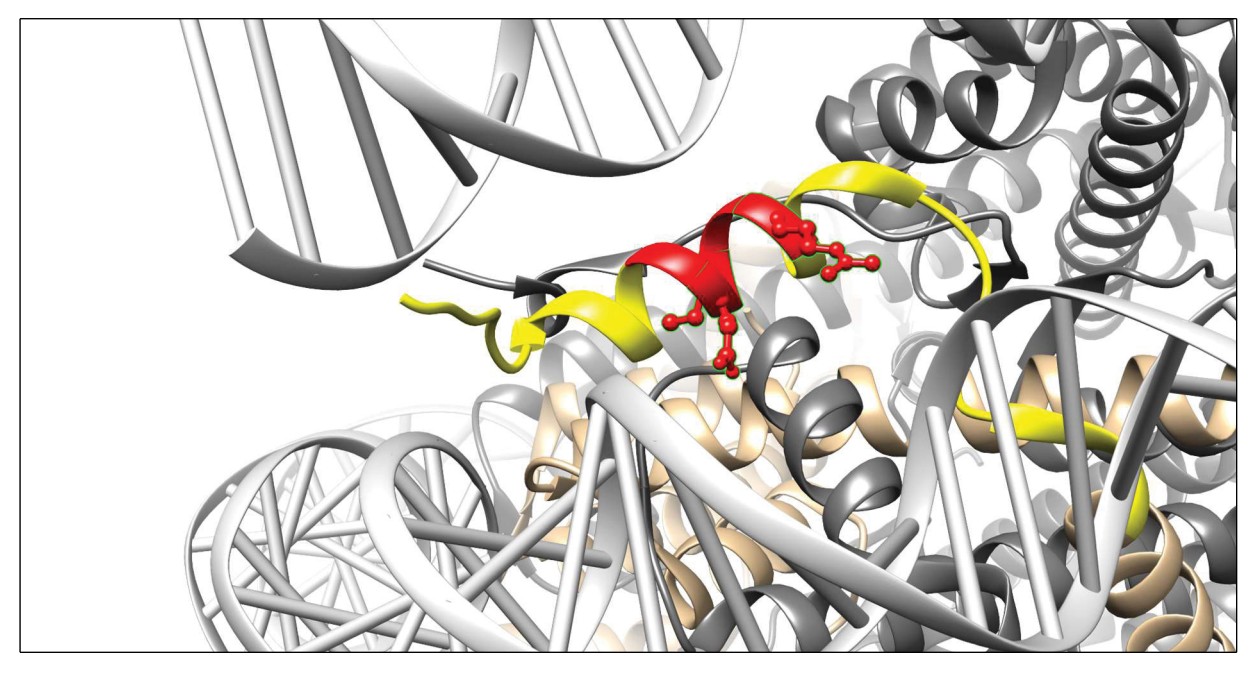

**Figure 4.** Visualisation of the 3D9 epitope in the nucleosome core complex. Visualization of the putative core epitope for 3D9 mapped on to histone H3 ribbon structure. The observed core binding site of 3D9 to the peptide arrays was depicted on histone H3 (light brown) in the nucleosome complex structure (file 3AZG.pdb). Part of the peptide sequences used in the peptide arrays is coloured in yellow. The core epitope (R)EIRR is displayed in red, with the atoms of the critical residues (Glu51, Ile52, and Arg54) shown. Binding profiles of antibody to linear and helical arrays, in addition to amino acid replacement analysis are presented in *Figure 4—figure supplements 1–3* and *Table 3*.

The online version of this article includes the following source data and figure supplement(s) for figure 4:

**Figure supplement 1.** Binding profiles recorded for 3D9 on the linear peptide array.

**Figure supplement 1—source data 1.** Linear and helical peptide epitope mapping.

**Figure supplement 2.** Binding profiles recorded for 3D9 on helical peptide mimics arrays.

**Figure supplement 2—source data 1.** Linear and helical peptide epitope mapping.

**Figure supplement 3.** Fine epitope mapping by replacement analysis.

**Figure supplement 3—source data 1.** Fine epitope mapping by replacement analysis.

anti-H3cit or anti-H2B antibodies stained relatively compact chromatin. Furthermore, colocalization analysis of 3D9 with H2B or with H3cit revealed that 3D9 was more commonly colocalised with H2B as compared to H3cit; overlap coefficients 0.463 (3D9-H2B) v 0.125 (3D9-H3cit), and 0.533 (3D9-H2B) v 0.122 (3D9-H3cit) for *Figure 10—figure supplement 1* and *Figure 11—figure supplement 1* respectively. A time course experiment in primary neutrophils stimulated with PMA also showed very rare occurrence of double, H3cit and 3D9, positive cells (*Figure 11—figure supplement 2*).

## Discussion

Decondensed chromatin is a defining feature of NETs. It occurs through PTMs that partially neutralise the histone positive charge and thus the affinity of histones for negatively charged DNA (*Papayannopoulos et al., 2010*; *Wang et al., 2009*). One way to achieve this is through proteolytic removal of the lysine and arginine rich histone tails. Using a biochemical and proteomic approach, we determined that H3 is cleaved at multiple sites in its N-terminal tail during the course of NET formation and notably, at a novel cleavage site in its globular domain, H3R49, at ~120 min. We exploited the specificity of this event to produce a mouse monoclonal antibody to the de novo histone H3 epitope, the new N-terminal beginning at R49. This antibody, 3D9, recognises human NETs induced by both microbial and host derived physiological stimuli and distinguishes netotic neutrophils from neutrophils

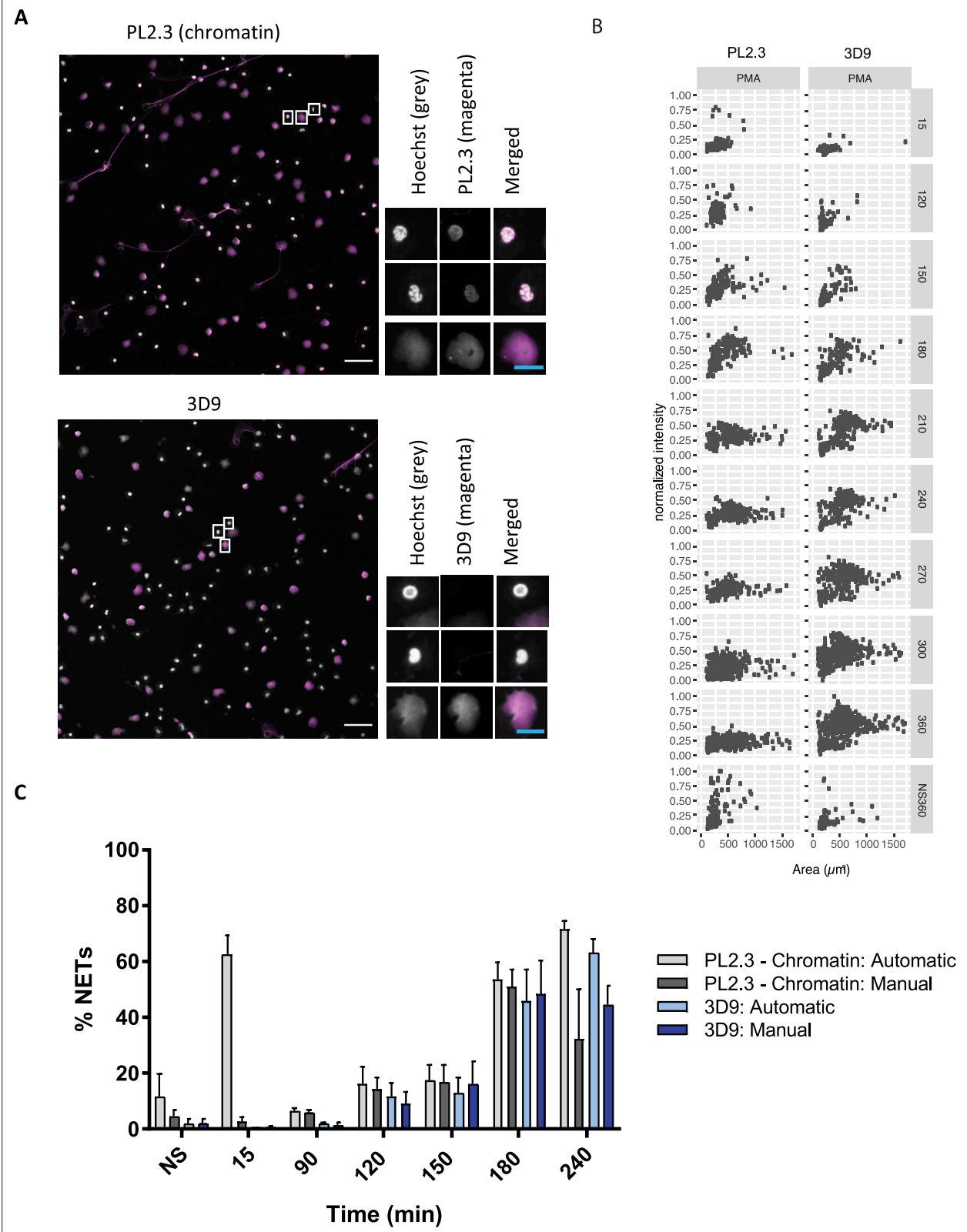

**Figure 5.** Comparison of NET quantification using an anti-chromatin antibody versus 3D9. (**A**) Confocal immunofluorescent microscopy of neutrophils stimulated with PMA (180 min) and stained with Hoechst and anti-chromatin antibody (PL2.3) or 3D9. Insets represent selected cells examined at higher magnification (63 x, Plan Apochromat, glycerol, numerical aperture 1.30) and presented as split channels in grayscale or merged as per the total field of view (20 x, Plan Apochromat, glycerol, numerical aperture 0.75). White scale bar - 50 μm, cyan scale bar - 10 μm. Images are representative of 3

*Figure 5 continued on next page*

*Figure 5 continued*

experiments (**B**) Comparison of the fluorescent distribution of PL2.3 versus 3D9 staining of PMA stimulated cells over time (6 h). Staining intensities were normalized over all images of the respective time course. NS:360: non-stimulated at 360 min. Analysis is performed on one data set that is representative of 3–4 independent time course experiments. Source data can be found in *Figure 5—source data 1* and *Figure 5—source data 2*. (**C**) Comparison of NET quantification using manual or automatic thresholding and segmentation procedures for chromatin antibody (PL2.3) and cleaved H3 antibody (or 3D9). Manual thresholding excludes cells/NETs with a weak signal whereas automatic thresholding includes all objects irrespective of signal. Images for analysis were taken using a upright fluorescent microscope. Graph represents the mean ± standard deviation, where n=3–5. Source data can be found in *Figure 5—source data 3*.

The online version of this article includes the following source data and figure supplement(s) for figure 5:

**Source data 1.** Characterisation of PL2.3 NET staining.

**Source data 2.** Characterisation of 3D9 NET staining.

**Source data 3.** Comparison of NET quantification using manual or automatic thresholding and segmentation procedures for chromatin antibody and 3D9.

**Figure supplement 1.** Workflow of NET analysis methods.

that die via apoptosis, necrosis and necroptosis, in vitro. It also discriminates between NETs and other cells in mixed blood cell fractions and, importantly, NETs in human histological samples.

Until now, histone citrullination is the only PTM that has been used for antibody-based detection of NETs. In this paper, we propose histone cleavage at H3R49 as a new histone PTM that can be used for detection of NETs from human samples. The use of H3cit for the detection of NETs is not without controversy, as discussed in depth by *Konig and Andrade, 2016*. Not all NETs are citrullinated at H3 and NET formation can occur in the absence or inhibition of citrullinating enzymes (*Kenny et al., 2017*). Like citrullination, not all NETs contain H3 cleaved at H3R49. With *Candida albicans*, some NET-like structures were not 3D9 positive (*Figure 6*). These may be remnants of other forms of cell death or they may be citrullinated NETs as has been shown by *Kenny et al., 2017*. By identifying the precise histone cleavage site through edmann degradation, we shed further light on this. The most commonly used H3cit antibody detects citrullination of R2, R8 and R17. However, histone cleavage at R49 would remove the H3cit epitopes, rendering these NET defining PTMs mutually exclusive on a single histone level. Indeed, co-staining by anti-H3cit and 3D9 in inflamed kidney, gallbladder, and appendix paraffin sections, and purified cells revealed extensive mutual exclusion of the two marks, even on a cellular level, and more abundant staining of decondensed chromatin by 3D9. Thus, we propose that 3D9 will allow detection of NETs induced by varied stimuli but that it may display a preference for more mature or proteolytically processed NETs or NETs that are citrullinated to a lesser degree or not at all. These findings may also suggest a need to re-evaluate NET studies derived from experiments using H3cit as the only method used to detect NETs.

In contrast to the present study, histone cleavage was reported as discriminating between NOX-dependent and NOX-independent pathways of NET formation (*Pieterse et al., 2018*). Using a sandwich ELISA approach, Pieterse et al concluded that, generally, the N-terminal histone tails are removed in NOX-dependent but not NOX-independent NET formation. They used a suite of N-terminal directed antibodies for H2B, H3, and H4, and all H3 epitopes were located N-terminal to the cleavage site H3R49. However, in the final biological sample testing, the authors did not examine H3. Interestingly, the authors also observed that, by immunofluorescent microscopy, all histone N-terminal antibodies failed to stain NETs at time points after cell lysis. Therefore, to us, this data that suggests that at later stages in NETosis the histone N-terminal tails, at least for H3, are cleaved irrespective of the pathway of activation. This is in line with our observations that both NOX-dependent (PMA) and NOX-independent (heme; nigericin) stimuli result in NETs that are recognised by 3D9 and supports our finding that histone H3 cleavage at R49 is a common feature in human NET formation.

In this study we detect NETs in fixed or denatured human samples from in vitro experiments and histological samples. While an ELISA with 3D9 revealed a preference for native NETs over isolated chromatin or recombinant H3 (under mild detergent conditions), immunoprecipitation of naive cell lysates by 3D9 also pulled down full length H3 as confirmed by immunoblot. It is not yet clear if this is due to co-immunoprecipitation due to the presence of low levels of clipped histone or if this represents true recognition of intact H3 by 3D9. Furthermore, a preliminary investigation revealed 3D9 reacts with a plasma protein(s) in ELISA and western blot - albeit of a higher molecular weight – (*Figure 3—figure supplement 4A*) and thus, care should be taken in the design of future assays

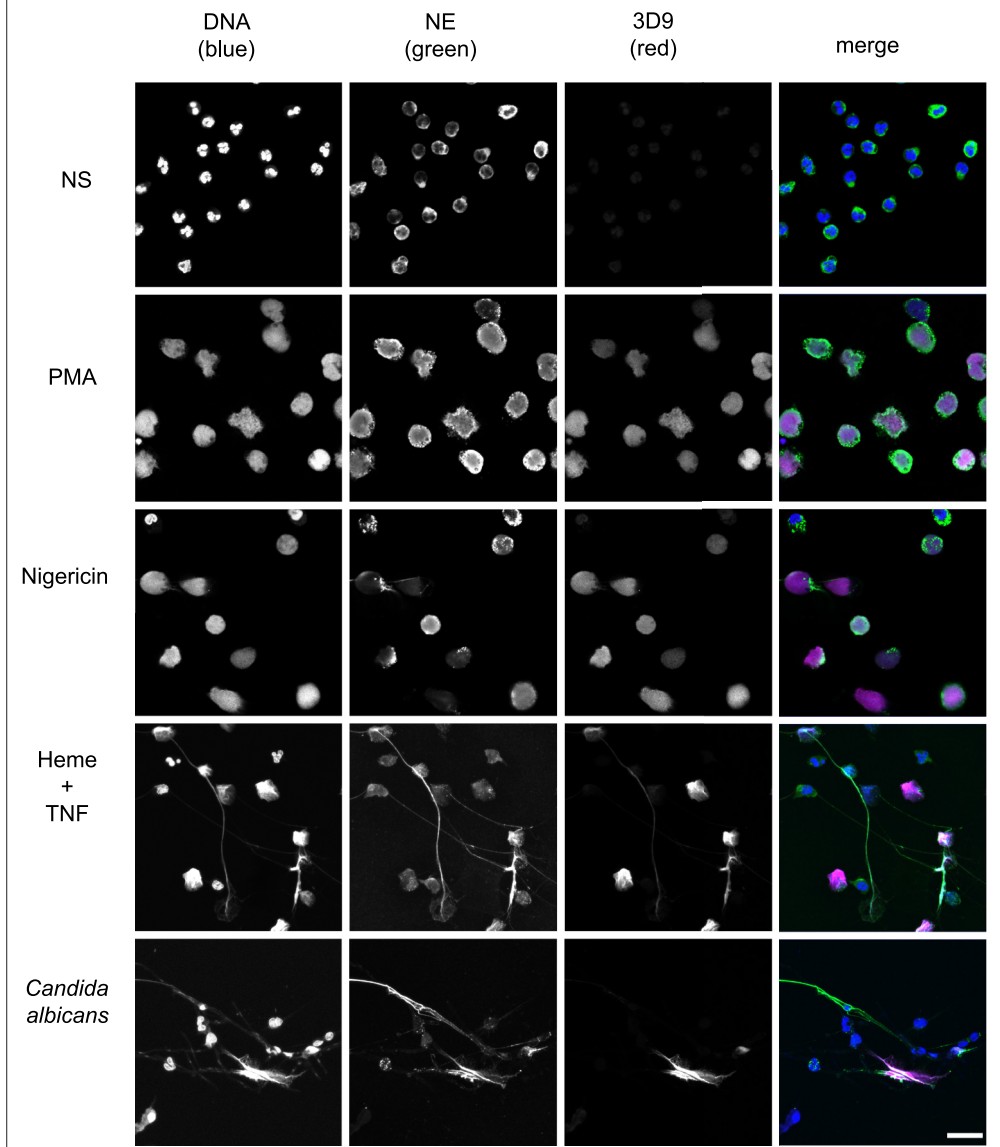

**Figure 6.** Detection by 3D9 of NETs from diverse stimuli. Immunofluorescence microscopy of neutrophils left unstimulated (NS), stimulated with PMA (50 nM, 2.5 hr), nigericin (15 µM, 2.5 hr), TNF primed and then stimulated with heme (20 µM, 6 hr), and neutrophils co-cultured with *Candida albicans* hyphae (MOI 5) for 4 hr. Samples were stained with Hoechst, anti-neutrophil elastase (NE) and 3D9. Scale bar – 50 µm. Images were taken on a confocal microscope at 20 x (Plan Apochromat, glycerol, numerical aperture 0.75) and are representative of three experiments with independent donors. Scale bar - 20 µm.

and selection of sample when detecting cleaved H3, NETs or vital NETs under native and mild detergent conditions. In particular, 3D9 alone may not be suitable for direct detection of NETs in complex biological fluids and a sandwich approach or colocalization with a neutrophil granule protein is likely to be critical.

More broadly, and applying to the general principles of NET detection, it is not yet possible to prove conclusively that the detected decondensed chromatin originates from the same cell source as the neutrophil proteins which decorate it. For example, in an infected necrotic wound to which to high numbers of neutrophils are recruited. Here, activated neutrophils might release both proteases (*Borregaard et al., 1993*) and citrullinating enzymes (*Spengler et al., 2015*; *Zhou et al., 2017*) that bind to and modify extracellular chromatin generating, according to the histological definition, a NET. This remains a conundrum that requires further exploration.

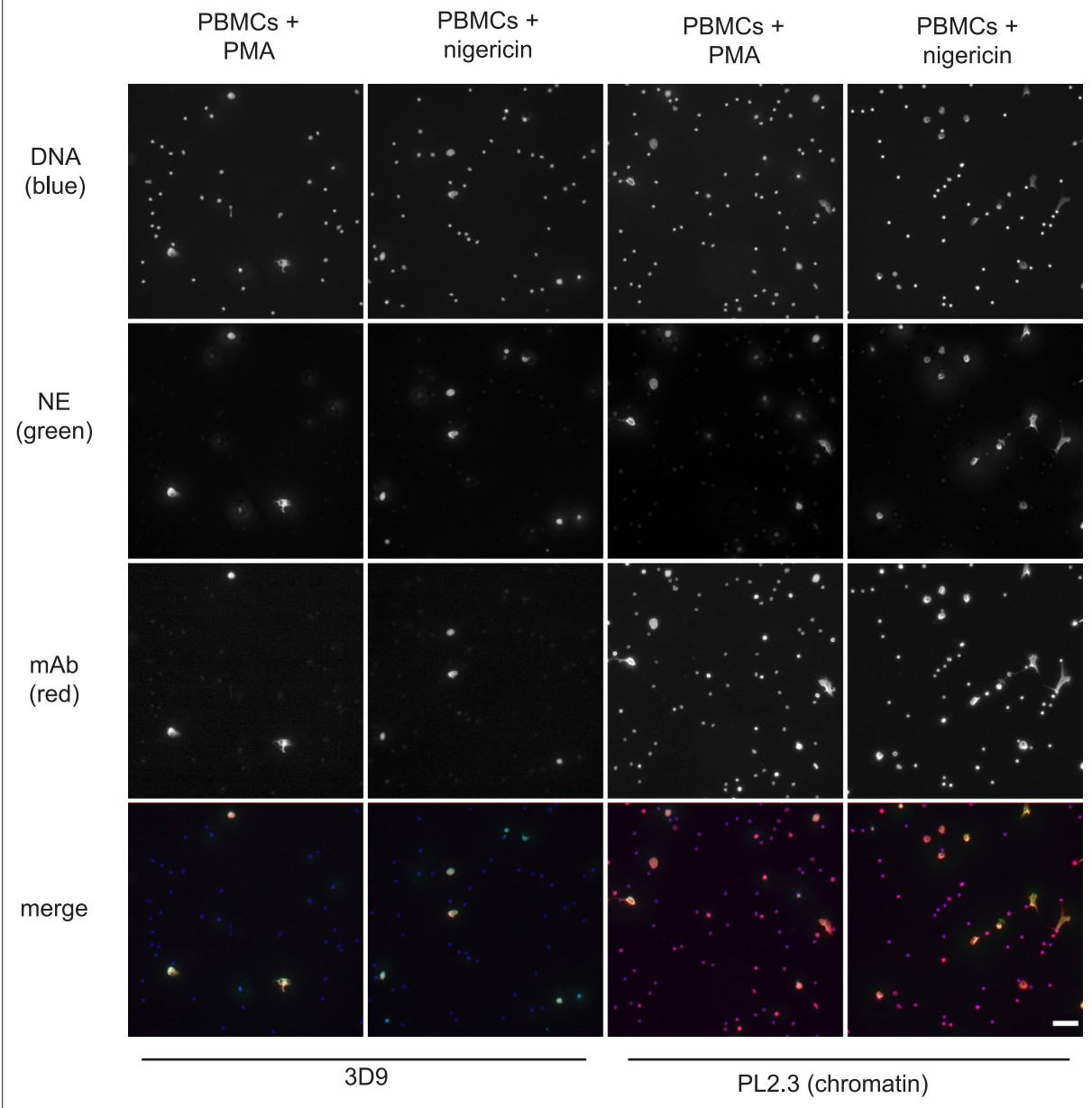

**Figure 7.** Detection of NETs in mixed cell fractions. Immunofluorescence microscopy of non-purified peripheral blood mononuclear cell (PBMC) fractions treated with the NET stimuli, PMA (50 nM, 2.5 hr) or nigericin (15 µM, 2.5 hr), and then stained with Hoechst, anti-neutrophil elastase (NE) and 3D9 or PL2.3. Images were taken on an upright fluorescent microscope at ×20 magnification (Fluotar, numerical aperture 0.50). The selected images are representative of three independent experiments. Scale bar – 50 µm.

N-terminal histone cleavage at H3R49 is a novel and so far undescribed cleavage site in any eukaryotic organism. Unusually, it is located in the globular rather than the unstructured tail region of H3. H3R49 is one of 6 key residues important for the regulation of H3K36me$^3$ and forms part of the structured nucleosome surface (*Endo et al., 2012*). Thus, we speculate that removal of the N-terminal tail, in its entirety, could lead to removal of higher order structure interactions and facilitate chromatin decondensation for example removal of H3K9me and its associated heterochromatin protein 1 interactions (*Jacobs and Khorasanizadeh, 2002*). To determine the contribution of histone cleavage at H3R49 to the process of chromatin decondensation, future work will focus on establishing the protease(s) responsible and the sequence of proteolytic events leading to this final truncation of H3 and NET formation. Given the specificity of this event and its restriction to NETotic forms of cell death, we propose that N-terminal cleavage at H3R49 is an example of histone 'clipping' in neutrophils – a

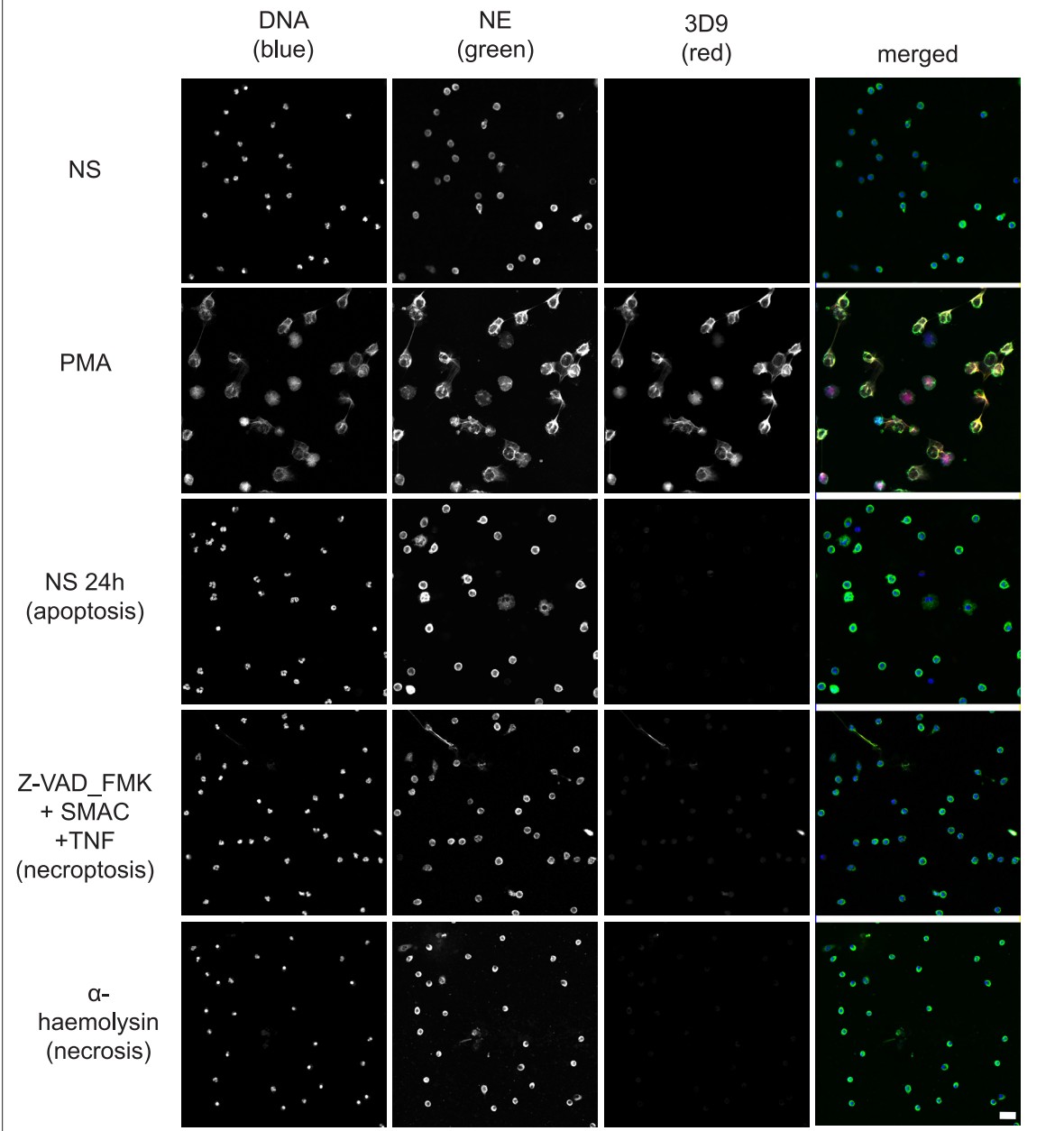

**Figure 8.** Comparison of 3D9 detection in response to apoptotic, necroptotic, and necrotic cell death stimuli. Confocal immunofluorescent microscopy of neutrophils stimulated with different cell death stimuli and subsequently stained with Hoechst, anti-neutrophil elastase (NE) and 3D9. NETs were induced with PMA (100 nM, 3 hr). Apoptosis was induced in resting neutrophils by incubation for 24 hr without stimulation overnight. Neutrophils were stimulated with Z-VAD-FMK (50 μM) plus SMAC mimetic (100 nM) plus TNF (50 ng/ml) for 6 hr to induce necroptosis. Necrosis was induced with the pore forming toxin α-haemolysin (25 μg/ml). Images were taken at 20 x (Plan Apochromat 20 x, numerical aperture 0.75) and are representative of three experiments. Scalebar 20 μm. A comparison was made with parallel samples stained with the chromatin antibody PL2.3 and are presented in *Figure 8— figure supplement 1*. Apoptotic samples were also stained for apoptotic markers and cleaved H3 (*Figure 8—figure supplement 2*).

The online version of this article includes the following figure supplement(s) for figure 8:

**Figure supplement 1.** Comparison of PL2.3 detection in response to apoptotic, necroptotic, and necrotic cell death stimuli.

**Figure supplement 2.** Staining of primary human neutrophils for markers of apoptosis and NETs.

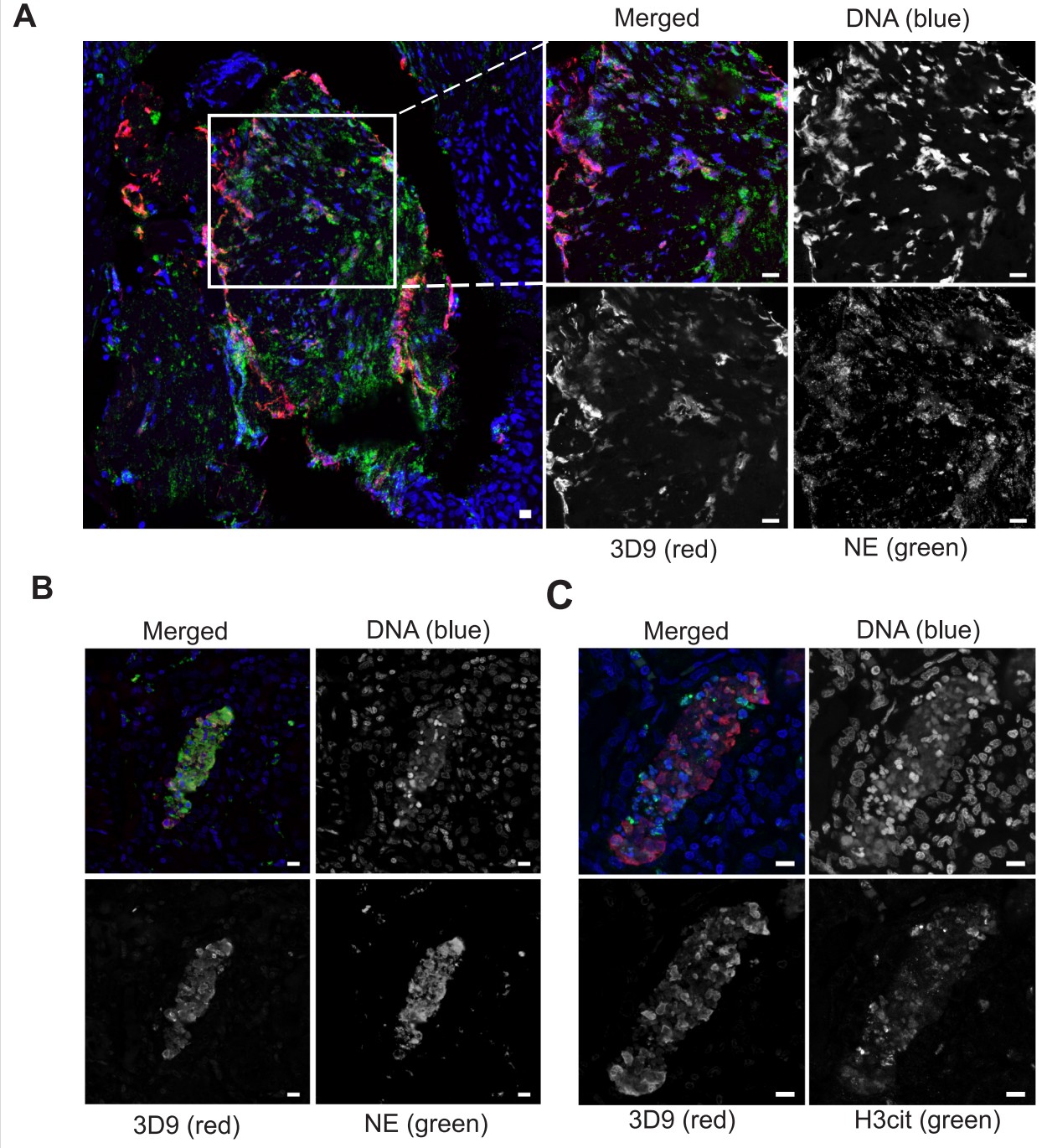

**Figure 9.** Detection of clipped histone 3 and NETs in human tissues. Paraffin embedded sections were stained with Hoechst, anti-NE and 3D9 or H3cit antibodies and examined by confocal microscopy at ×63 magnification (Plan Apochromat, glycerol, numerical aperture 1.30). Scale bar - 10 µm (**A**) human tonsil, denoted 'normal' by commercial provider but showing infiltration of neutrophils demonstrating an inflammatory event. (**B**) & (**C**) human kidney, denoted 'inflammed' by commercial provider.

The online version of this article includes the following figure supplement(s) for figure 9:

**Figure supplement 1.** Hematoxylin and eosin (HE) stain of kidney section.

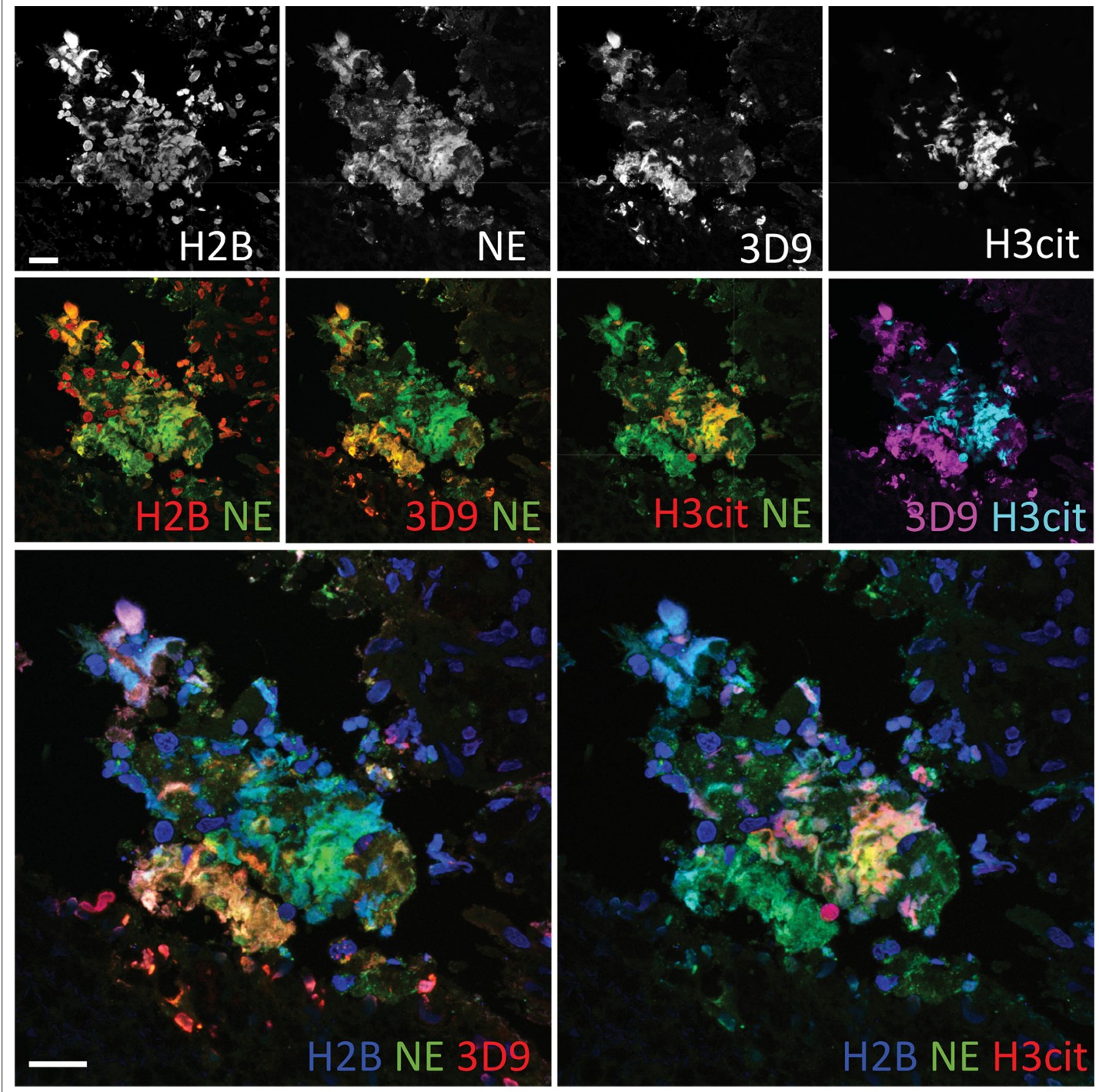

**Figure 10.** Comparison of Clipped H3, H3cit & H2B staining in the gallbladder from an appendicitis patient. Paraffin embedded sections were stained with Hoechst, anti-NE and histone antibodies 3D9, H3cit and H2B and examined by confocal microscopy at ×63 magnification (Plan Apochromat, glycerol, numerical aperture 1.30). Scale bar – 20 μm.

The online version of this article includes the following figure supplement(s) for figure 10:

**Figure supplement 1.** Hematoxylin & eosin (HE) stain of gallbladder and colocalization analysis.

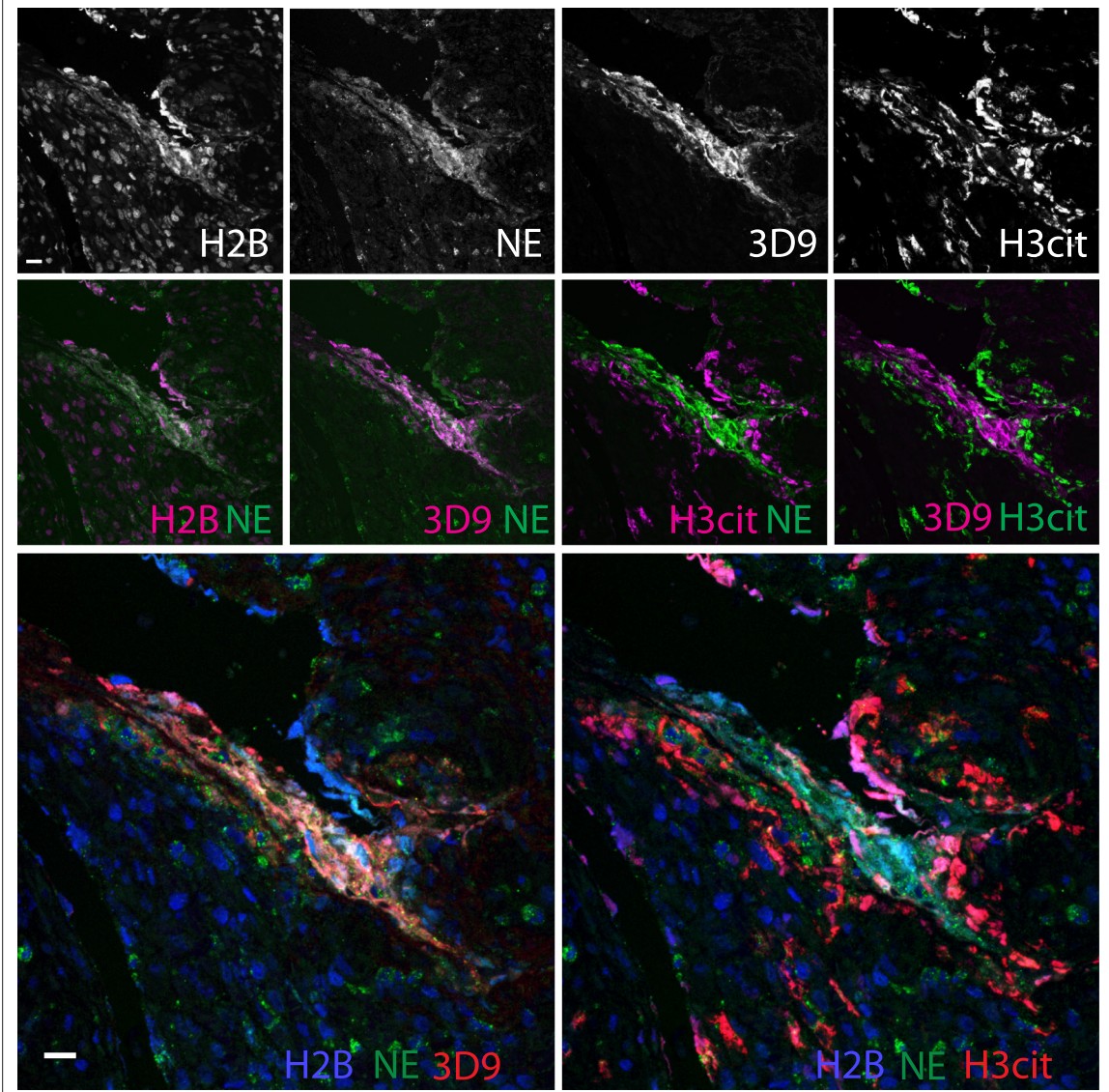

**Figure 11.** Comparison of Clipped H3, H3cit, and H2B staining in the appendix of an appendicitis patient. Paraffin embedded sections were stained with Hoechst, anti-NE and 3D9 or H3cit antibodies and examined by confocal microscopy at ×63 magnification (Plan Apochromat, glycerol, numerical aperture 1.30). Scale bar - 20 µm.

The online version of this article includes the following figure supplement(s) for figure 11:

**Figure supplement 1.** Hematoxylin and eosin stain of inflamed appendix and colocalization analysis.

**Figure supplement 2.** Time course of 3D9 and H3cit co-staining of primary human neutrophils stimulated with PMA.

term proposed by the histone/histone proteolysis field for specific histone cleavage sites for which a biological function has been demonstrated (*Dhaenens et al., 2015*).

## Limitations of the study

While this study has attempted to characterise 3D9 as tool to detect human NETs, it is limited by the range of NET inducing stimuli tested. In particular, we were unable to produce NETs in response to immune complexes and it will be important to examine the usefulness of 3D9 in future studies into aberrant NET production in the pathophysiology of autoimmune disease among others. Critically, its usefulness with patient serum samples will need to be demonstrated, bearing in mind the cross reaction with a plasma protein. While here we compared 3D9 staining to another NET detection method - NE and chromatin antibodies - in vitro, and included another NET surrogate marker - H3cit

- in tissues, a more in depth cross-comparison between 3D9, H3cit and other NET staining methods will be needed to fully understand the NET landscape and usefulness of 3D9 for tissue samples going forward. Giving consideration to the caveats that have arisen for the use of H3cit, and that H3cit may not be detecting NETosis in all instances, it will be important to identify the protease responsible for cleavage at H3R49 and, in turn, examine chromatin and cell death responses in cells expressing this protease to help fully establish the specificity of H3R49 clipping in NETosis.

## Conclusion

This study represents the first identification of a distinctive and, so far, exclusive marker of NETs and describes the development and characterisation of a complementary antibody to facilitate easier detection of human NETs. Analogous to finding a smoking gun at a crime scene, the monoclonal antibody 3D9 detects evidence of the proteolytic events that occur in NETosis – the proteolytic signature, histone cleavage at H3R49. In doing so, 3D9 discriminates NETs from chromatin of other cells and chromatin of neutrophils that die via alternative mechanisms. This added layer of specificity will simplify the detection of NETs in tissue samples and facilitate comparison of quantitative studies between labs. This will be an important step in assessing the contribution of extracellular chromatin and NETs to disease pathology.

# Materials and methods

**Key resources table**

| Reagent type (species) or resource | Designation | Source or reference | Identifiers | Additional information |
|---|---|---|---|---|
| Cell line (human) | A549 human lung epithelial cells; A549 | American Type Cell Collection | ATCC:CCL-185 | |
| Genetic reagent (bovine) | Calf thymus DNA; purified DNA | Invitrogen | | |
| Antibody | Rabbit polyclonal anti-histone H3 C-terminal; anti-H3-C; αH3-C | Active Motif | Active Motif:61277 | Western blot (1:15000) |
| Antibody | Rabbit polyclonal Histone H4 C-terminal; αH4-C | Abcam | Abcam:ab10158 | Western blot (1:5000) |
| Antibody | Rabbit monoclonal anti-histone H4 N terminal | Upstate Millipore, Sigma Aldrich | SigmaAldrich:05–858 | Western blot (1:30000) |
| Antibody | Rabbit monoclonal anti-GAPDH; GAPDH | Cell Signalling Technology | CST:2118 | Western blot (1:5000) |
| Antibody | Mouse monoclonal 3D9; 3D9: mouse anti-cleaved histone 3 3D9 | This paper | | Custom synthesis by Genscript. Western blot (1 µg/ml), immunofluorescence (1–2 µg/ml) |
| Antibody | Rat monoclonal anti-histone H3 N-terminal; anti H3-N; αH3N | Active Motif | Active motif:61647 | Western blot (1:15000) |
| Antibody | Mouse monoclonal anti-H2A-H2B-DNA; PL2.3; anti-chromatin | PMID:1371530 | | Immunofluorescence (1 µg/ml) |
| Antibody | Rabbit polyclonal anti neutrophil elastase; anti-NE; αNE | Calbiochem | Calbiochem 481001 | Immunofluorescence (1:500) |
| Antibody | Rabbit polyclonal anti-histone H3 (citrullinine R2+R8+R17); anti- H3cit | Abcam | Abcam:ab5103 | Immunofluorescence (1 µg/ml) |
| Antibody | Chicken polyclonal anti-histone H2B | abcam | Abcam:ab134211 | Immunofluorescence (1:400) |
| Antibody | Polyclonal Sheep anti-ELANE: anti-NE | LSBio | LSBio:LS-B4244-50 | Immunofluorescence (5 µg/ml) |
| Antibody | Rabbit polyclonal Anti-mouse Bridging antibody | Active Motif | Active motif:53017 | (50 µg) |
| Antibody | Mouse monoclonal (1H7A8) IgG1, k, isotype control | Genscript | | Provided by Genscript as an isotype control (1 µg/ml) |

*Continued on next page*

*Continued*

| Reagent type (species) or resource | Designation | Source or reference | Identifiers | Additional information |
|---|---|---|---|---|
| Peptide, recombinant protein | Recombinant human histone H3.1 | New England Biolabs | NEB:M2503S | |
| Peptide, recombinant protein | Peptide H-REIRRK(RREIR)C-NH2; immunising peptide REIRR | This paper | | Custom synthesis by Eurogentec |
| Peptide, recombinant protein | Peptide H-AARKSK(SKRAA)C-NH2; negative screening peptide | This paper | | Custom synthesis by Eurogentec |
| Peptide, recombinant protein | Peptide H-REIRRK (RRIER_NH2; competition peptide) | This paper | | Custom synthesis by Eurogentec |
| Peptide, recombinant protein | Peptide H-TGGVKK(KVGGT)-NH2; negative control peptide | This paper | | Custom synthesis by Eurogentec |
| Commercial assay or kit | BD OptEIA TMB substrate reagent set | BD Biosciences | RRID:AB_2869044 | |
| Commercial assay or kit | Picogreen Assay kit; Quant-iT PicoGreen dsDNA Assay Kits and dsDNA Reagents | Thermofisher Scientific | Thermofisher:P7589 | |
| Commercial assay or kit | Cytoxicity assay; CytoTox 96 Non-Radioactive Cytotoxicity assay | Promega | Promega:G1780 | |
| Chemical compound, drug | Hoechst DNA stain (33342) | Invitrogen Molecular Probes | Invitrogen:H1399 | Immunofluorescence 1 µg/ml |
| Chemical compound, drug | Neutrophil elastase inhibitor GW311616A; NEi | Biomol | Biomol:Cay27957-1 | |
| Chemical compound, drug | AEBSF hydrochloride; peflabloc | Millipore, Sigma Aldrich | Sigma Aldrich:124839 | |
| Chemical compound, drug | Cathepsin G inhibitor I; CGi | Calbiochem, Sigma Aldrich | Sigma Aldrich:219372 | |
| Software, algorithm | Fiji; Image J | https://doi.org/10.1038/nmeth.2019 | | |
| Software, algorithm | NETalyser | This paper | | (Scripts deposited at https://github.com/tulduro/NETalyser) |
| Software, algorithm | Volocity | Perkin Elmer | | |

## Reagents

All reagents were purchased from common vendors of laboratory reagents for example Sigma Aldrich or VWR Deutschland unless otherwise stated.

## Blood collection and ethical approval

Venous blood was collected from healthy donors who had provided informed consent according to the Declaration of Helsinki. Ethical approval was provided by the ethics committee of Charité-Universitätsmedizin Berlin and blood was donated anonymously at Charité Hospital Berlin.

## Purification and culture of human peripheral blood neutrophils

Neutrophils were isolated as described by *Amulic et al., 2017*. Briefly, venous whole blood was collected in EDTA and separated by layering over equal volume Histopaque 1119 and centrifugation at 800 *g* (20 min). The pinkish neutrophil rich fraction was collected and washed once by the addition of 3 volumes of wash buffer (PBS, without $Mg^{2+}$ or $Ca^{2+}$ [Gibco] supplemented with 0.5% [w/v] human serum albumin [HSA, Grifols]) and centrifugation at 300 *g* (10 min). The neutrophil fraction was further purified by density gradient centrifugation - Percoll (Pharmacia) gradient from 85% to 65% (v/v). Purified cells were collected from the 80% to 70% fractions and washed once before being resuspended in wash buffer. Cells were counted using a CASY cell counter.

For all experiments, unless indicated, neutrophils were cultured RPMI (GIBCO 32404014) supplemented with 10 mM HEPES and 0.1% (w/v) HSA, which had been preequilibrated in $CO_2$ conditions

for 1 hr. For some stimuli, the HSA content was reduced to 0.05% or 0% HSA as indicated in the figure legends. Cells were routinely cultured at 37 °C, 5% $CO_2$ unless indicated. For all experiments, stimuli were added to cell reactions as 10 X working stock solutions freshly diluted in RPMI. For inhibition experiments, a 10 X inhibitor stock and appropriate vehicle controls, were added to the cells and preincubated for the times stated in the figure legends.

## Cytotoxicity assay

To assess any toxicity of inhibitors LDH release was assessed using the CytoTox 96 Non-Radioactive Cytotoxicity Assay (Promega), according to the manufacturer's instructions. Neutrophils were seeded in RPMI in a 6 well plate with 1x105 cells/100 µl. Inhibitors were preincubated with cells for 30 min at 37 °C at concentrations indicated in the figures. Assay was performed in triplicate.

## Reactive oxygen species assay

Reactive oxygen species (ROS) production was measured using a luminol-HRP assay as described by *Amulic et al., 2017*. Additionally, as all stages of the assay were performed in atmospheric conditions, normal tissue culture media was replaced with a carbonate and phenol red-free RPMI supplemented as before (Seahorse XF RPMI Medium #103336–100, Agilent). Inhibitors were preincubated with cells for 30 min at 37 °C at concentrations indicated in the figures. Immediately before stimulation, luminol (50 µM) and HRP (1.2 U/ml) were added followed by 50 nM PMA. Luminesce measurements were taken every 30 s using a luminescence plate reader (VICTOR Light luminescence counter, Perkin Elmer) and expressed as Relative Light Units (RLU). Assay was performed in triplicate.

## Neutrophil and NET lysate preparation

To analyse proteins, lysates were prepared from stimulated or resting neutrophils. Cells were seeded in culture medium in 1.5 ml microcentrifuge tubes at $1x10^7$ cells/ml with $5x10^6$ cells per time point. After addition of the inhibitor or agonist, cells were gently mixed and incubated at 37 °C with gentle rotation. At the specified time points, protease inhibitors - 1 mM AEBSF, 20 µM Cathepsin G inhibitor I (Calbiochem), 20 µM neutrophil elastase inhibitor GW311616A (Biomol), 2 X Halt protease inhibitor cocktail (PIC, Thermofisher Scientific), 10 mM EDTA, 2 mM EGTA - were added directly to the cell suspension. Cells were gently mixed and centrifuged at 1000 *g* (30 s) to collect all residual liquid. Freshly boiled 5 X sample loading buffer (50 mM Tris-HCl pH 6.8, 2% [w/v] SDS, 10% glycerol, 0.1% [w/v] bromophenol blue, 100 mM DTT) was added to samples which were then briefly vortexed and boiled (98 °C) for 10 min with agitation and flash frozen in liquid nitrogen for storage at –80 °C.

## 1D SDS-PAGE and immunoblot blot

For routine protein analysis, samples were analysed by 1D SDS-PAGE and immunoblotted. Samples were thawed on ice, boiled at 98 °C (10 min) and sonicated to reduce viscosity (Braun sonicator, 10 s, cycle 7, power 70%). Proteins were applied to NuPAGE 12% gels (Invitrogen, Thermofisher) and run at 150 V in MES buffer (Thermofisher Scientific). Proteins were transferred by western blot onto PVDF (0.2 µm pore size, Amersham GE Healthcare) using the BioRad wet transfer system (buffer: 25 mM Tris, 192 mM glycine, 20% methanol, protocol: 30 min at 100 mA, 120 min at 400 mA). Blotting efficacy was assessed by Ponceau S staining. Blots were blocked with TBST (TBS pH 7.5, 0.1% [v/v] Tween-20) with 5% [w/v] skimmed milk, for 1 hr at RT. Blots were then incubated with the following primary antibodies overnight at 4 °C or for 2 hr at RT: rabbit anti-histone H3 C-terminal pAb, 1:15000 (Active motif #61277); rat anti-histone H3 N-terminal mAb, 1:1000 (Active Motif #61647, aa 1–19); Histone H4 C-terminal, 1:5000 (Abcam 10158 – aa 50 to C terminal); rabbit anti-histone H4 N-terminal mAb, 1:30,000 (Upstate, Millipore #05–858, aa17-28); rabbit GAPDH mAb, 1:5000 (Cell Signalling Technology, #2118); mouse 3D9 1 µg/ml (produced in this study) – all diluted in TBST with 3% (w/v) skimmed milk. After washing with TBST (3x5 min), blots were blocked for 15 min as before and then probed with secondary HRP conjugated antibodies (Jackson ImmunoResearch -diluted 1:20,000 in 5% skimmed milk TBST) for 1 hr at RT. Blots were washed in TBST (3x5 min) and developed using SuperSignal West Dura Extended Duration Substrate (ThermoFisher Scientific) and an ImageQuant Gel imager (GE Healthcare).

## Immunofluorescent staining of in vitro samples

For immunofluorescent imaging of purified cells, neutrophils/PBMCs were seeded in 24 well dishes containing glass coverslips, with $1x10^5$ cells per well and incubated at 37 °C for 1 hr to allow to adhere

to the coverslip. At this stage, inhibitors and priming factors were included as indicated. Reactions were stopped by the addition of paraformaldehyde (2% [w/v]) for 20 min at RT or 4 °C overnight. After fixation, cells were washed and stained as previously described (*Brinkmann et al., 2012*). Briefly, all steps were performed by floating inverted coverslips on drops of buffer on laboratory parafilm. Cells were permeabilised with PBS, pH 7.5, 0.5% (v/v) Triton X-100 for 3 min. For screening and quantification experiments, this permeabilization step was extended to 10 min. Samples were washed (3x5 min) with PBS, 0.05% (v/v) Tween 20 and incubated with blocking buffer - PBS pH 7.5, 0.05% (v/v) Tween 20, 3% (v/v) normal goat serum, 3% (w/v) freshwater fish gelatin, 1% (w/v) BSA - for 20 min at RT and then probed with primary antibodies diluted in blocking buffer and incubated overnight at 4 °C. Primary antibodies: anti-chromatin (H2A-H2B-DNA complex) mouse mAb PL2.3, 1 µg/ml *Losman et al., 1992*; neutrophil elastase, rabbit pAb 1:500 (Calbiochem); mouse serum for screening, 1:100; hybridoma supernatants, neat; 3D9 mouse mAb, 1 µg/ml. PL2.3 colocalisation with anti-NE was used as a control for NET detection in all in vitro experiments. Samples were then washed as before. Alexa labelled secondary antibodies (Invitrogen) were diluted 1/500 in blocking buffer and incubated for 2 hr at RT. DNA was stained with Hoescht 33342 (Invitrogen, Molecular Probes) 1 ug/ml, incubated with the secondary antibody step. Samples were washed in PBS followed by water and mounted in Mowiol mounting medium.

## Histone extraction from neutrophils

Histone enriched fractions were prepared from resting neutrophils and NETs according to a method modified from *Shechter et al., 2007*. Neutrophils ($4–8x10^7$) were resuspended in 13 ml of RPMI (without HSA) in a 15 ml polypropylene tube and incubated on a roller at 37 °C with PMA 50 nM for 90 min. After stimulation, 1 mM AEBSF was added to inhibit further degradation by NSPs and cells were cooled on ice for 10 min. All subsequent steps were performed on ice or 4 °C where possible. Cells and NETs were pelleted by centrifugation at 1000 $g$, 10 min. Samples were resuspended in ice-cold hypotonic lysis buffer (10 mM Tris-HCl pH 8.0, 1 mM KCL, 1.5 mM $MgCl_2$, 1 mM DTT supplemented with protease inhibitors just before use – 1 mM AEBSF, 20 µM NEi, 20 µM CGi, 2 X PIC, 10 mM EDTA) using 1 ml of buffer/$5x10^6$ cells. Cells were then incubated at 4 °C on a rotator for 30 min before being passed through a syringe to aid lysis and shearing of intact cells. Nuclei and NETs were collected by centrifugation at 10,000 $g$, 10 min, discarding the supernatant. To disrupt nuclei, samples were resuspended in $dH_2O$ (1 ml/$1x10^7$ cells) supplemented with protease inhibitors, as before, and incubated on ice for 5 min with intermittent vortexing. NP40 (0.2% [v/v]) was added to help lysis and disruption of NETs and samples were sonicated briefly (10 s, mode 7, power 70%). To extract histones, $H_2SO_4$ (0.4 N) was added to samples, and vortexed briefly. Samples were then incubated, rotating, for 2–3 h. Histone enriched fractions were collected by aliquoting samples into multiple 1.5 ml microcentrifuge tubes and centrifuging at 16,000 $g$ for 10 min, followed by collection of the supernatants. To minimise further processing of histones, proteins were immediately precipitated overnight by dropwise addition of trichloroacetic acid to a final concentration of 33% followed by mixing. The next day precipitated proteins were pelleted by centrifugation at 16,000 $g$, 10 min. The supernatants were discarded and waxy pellets were washed once with equal volume ice-cold acetone with 0.2% (v/v) HCl and 5 times with ice-cold acetone alone. Pellets were allowed to air dry for 5 min before being resuspended with 1 ml (per $5x10^6$ cells) of $dH_2O$ plus 1 mM AEBSF. For difficult to resuspend pellets the mixture was vigorously shaken at 4 °C overnight before samples were centrifuged, as before, to remove undissolved protein. Pooled supernatants for each sample were lyophilised and stored at –80 °C until histone fractionation.

## Purification of histone H3

Histones were fractionated by RP-HPLC according to the method described by *Shechter et al., 2007*. After lyophilisation samples were resuspended in 300 µl Buffer A (5% acetonitrile, 0.1% trifluoroacetic acid) and centrifuged at 14,000 $g$ to remove particulate matter. 150 µl of clarified sample was mixed with 40 µl Buffer A before being applied to a C18 column (#218TP53, Grave Vydac) and subjected to RP-HPLC (Waters 626 LC System, MA, US) as described by *Shechter et al., 2007*. The flow rate was set to 1 ml min$^{-1}$ and fractions were collected at 30 s intervals from minute 30–55. All fractions were lyophilised and stored at –80 °C until analysis. To determine which fractions contained H3 and cleaved species, each fraction was dissolved in 50 µl $dH_2O$ and 5 µl was subjected to SDS-PAGE and

either stained with Coomassie blue stain or transferred to PVDF and immunoblotted for H3 and H4 as described already.

## Two-dimensional electrophoresis (2-DE) of purified histones

To determine the cleavage sites, H3 containing fractions were pooled and subjected to a small gel 2-DE procedure (*Jungblut and Seifert, 1990*). Briefly, pooled fractions were denatured in 9 M urea, 70 mM DTT, 2% Servalyte 2–4 and 2% Chaps. Samples (30 µl) were applied to 1.5-mm-thick isoelectric focusing (IEF) gels using ampholytes 7–9 and a shortened IEF protocol was used: 20 min 100 V, 20 min 200 V, 20 min 400 V, 15 min 600 V, 5 min 800 V, and 3 min 1000 V, (a total of 83 min and 500 Vh) in 8 cm long IEF tube gels. Separation in the second dimension was performed in 6.5 cm x 8.5 cm x 1.5 mm SDS-PAGE gels. Duplicate gels were prepared; one stained with Coomassie Brilliant Blue R250 for excision of spots for mass spectrometry identification; and the second transferred to PVDF as follows. Proteins were blotted onto PVDF blotting membranes (0.2 µm) with a semidry blotting procedure (*Jungblut et al., 1990*) in a blotting buffer of 100 mM borate. Spots were stained by Coomassie Brilliant Blue R250 and analysed by N-terminal Edman degradation sequencing (Proteome Factory, Berlin, Germany).

## Mass spectrometry

Protein spots were excised from 2D gels and transferred to 0.5 ml Eppendorf tubes. Samples were destained in 500 µl of 200 mM ammoniumbicarbonate (ABC) in 50% acetonitrile (ACN) for 30 min at 37°C and equilibration in 200 µl of 50 mM ABC, 5% ACN for 30 min at 37°C. The samples were then dried at room temperature for 60 min. Protein digestion was performed overnight at 37 °C with 100 ng trypsin (spots 1, 3 ,7 with 200 ng) in 25 µl of 50 mM ABC, 5% ACN. The supernatants were transferred into new Eppendorf tubes and additional peptides were extracted by applying 25 µl of 60% ACN, 0.5% trifluoroacetic acid (TFA) for 10 min, followed by 25 µl of 100% ACN for 10 min. All supernatants were combined and dried in an Eppendorf Concentrator at 45 °C. After solubilization in 15 µl of 0.1% TFA, the peptides were desalted and concentrated with ZipTips and eluted with 1 µl alpha-cyano-4-hydroxycinnamic acid (5 mg/ml in 60% ACN, 0.3% TFA) onto the MALDI plate. Peptide mass fingerprints (PMF) and fragment spectra (MSMS) of the five most intense peaks were measured using a 4700 Proteomics Analyzer (AB Sciex). The database search was performed applying the Mascot MS/MS Ion Search function for searching the Swiss-Prot subset of human proteins. A peptide mass tolerance of 30 ppm and ± 0.3 Da for the fragment mass tolerance was allowed. One missed cleavage, oxidation of methionine, N-terminal acetylation of the protein, propionamide at cysteine residues and N-terminal pyroglutamic acid formation were defined as variable modifications. The following identification criteria were used: minimum 30% sequence coverage; or minimum 15% sequence coverage and one MS/MS confirmation; or sequence coverage below 15% and at least two MS/MS confirmations.

## Antibody generation

Immunising and screening peptides are outlined in *Table 3* and were synthesized by Eurogentec (Belgium). A portion was further conjugated to Key Lymphocyte Haemoglutinin (KLH) for immunization. Immunisation of mice, preliminary ELISA screening and production of hybridomas were performed by Genscript as follows. Six mice (3 x Balb/c and 3x C57/BL6) were immunized with branched peptides. Mice were bled and effective immunization was assessed using a direct ELISA. The ELISA and subsequent inhouse immunoblot and immunofluorescent microscopy screening strategy are outline in *Figure 3—figure supplement 1*. Following selection of effectively immunized animals, a further boost injection of the immunogen was given before isolation of spleen cells for hybridoma production. The resulting hybridoma supernatants were screened similarly. Large-scale culture of supernatants and purification of antibodies was performed by Genscript.

## ELISA

To assess 3D9 specificity for NETs, 3D9 was used in an indirect ELISA to detect cleaved H3 in purified NETs, chromatin, recombinant H3 and DNA. NETs were prepared by seeding $3x10^6$ neutrophils in a six-well dish and incubating for 3–6 hr with 100 nM PMA. NETs were gently washed 3 times with equal volume PBS, before being collected in 300 µl PBS. Clumped NETs were disrupted by sonicating briefly

(3 s, mode 7, power 70% - Braun Sonicator) and then snap frozen and stored at –80 °C. Chromatin was prepared from lung epithelial cells (A549) as previously described by *Shechter et al., 2007*. The final nuclear pellet was resuspended in dH$_2$O and sonicated as before and stored at –80 °C. The DNA content of NETs and chromatin was assessed by PicoGreen assay according to the manufacturer's instructions (Thermofisher Scientific). Beginning at 1 µg/ml (of DNA content), serial dilutions of NETs, chromatin and calf thymus DNA (Invitrogen) were prepared in lo-DNA bind Eppendorf microcentrifuge tubes. A similar dilution series of recombinant histone H3 (New England Biolabs) was prepared starting at 1 µg/ml protein. All dilutions were performed in PBS. One hundred microliters of each sample, in duplicate, at dilutions 1 µg/ml to 1 ng/ml, was aliquoted in a Nunc Maxisorb 96 well dish and immobilised overnight at 4 °C, 250 rpm. The immunising peptide, REIRR (10 ng/ml) was used as a positive control. The following day all wells were washed 6 times with wash buffer (PBS, 0.05% Tween 20) and then blocked with 200 µl of blocking solution (1% BSA in wash buffer) for 2 hr (RT). Wells were washed once with wash buffer and incubated with 100 µl of 3D9 (2 µg/ml, in blocking solution) at RT (2 hr) with gentle shaking (250 rpm). Wells were washed 6 times as before and then incubated with 100 µl of secondary HRP conjugated anti-mouse (Jackson laboratories) at 1:100,000 dilution in blocking solution and incubated as before. Finally, wells were washed 6 times as before and HRP activity was detected using TMB (3,3', 5,5' tetramethylbenzidine) reagent (BD OptEIA) according to the manufacturer's instructions (incubating for 15–30 min). The assay was stopped by the addition of 100 µl H$_2$SO$_4$ (0.16 M) and absorbance (450 nm) was measured on a 96-well plate reader (VERSAmax, Molecular Devices, CA, US).

The same approach was used to examine 3D9 interactions with immobilised serum and plasma proteins. Plasma was isolated from whole blood collected with S. Monovette sodium citrate tubes (Sarstedt). Whole blood was centrifuged at low speed to minimise cell lysis (150 xg, 20 min with no brake). Prostaglandin E1 (1 µm) was added to inhibit platelet activation and samples were further centrifuged at 650 xg (8 min) to collect cell free plasma and further centrifuged at 2000 xg (10 min) before being aliquoted and stored at –80 °C. Serum was isolated by collection of whole blood in S. Monovette serum tubes (silicate clotting activator) and incubation with gentle rotation for 30 min at RT before centrifugation at 2000 xg at 4 °C (10 min) and collection of serum.

## Immunoprecipitation (IP) of clipped histones

Magnetic Protein G coupled beads (Invitrogen) were washed (x3) with equal volume PBS. For each IP sample 20 µl of beads were prepared. As 3D9 was a mouse IgG1, a bridging antibody (anti-mouse, Active motif #53017) was used to improve binding to protein G couple beads. Bridging antibody (50 µg) was incubated with 20 µl of magnetic beads for 1 hr at 4 °C with gentle inversion. Beads were washed twice with PBS. Beads were then incubated with 50 µg 3D9 or isotype control (IgG1, k - Genscript clone 1H7A8) for 3 hr at RT (with gentle rotation). Beads were washed x3 and resuspended in 2 ml of PBS in a 15 ml falcon tube. To permanently crosslink the antibodies, PFA fixation was used. 2 ml of 1% PFA (in PBS) was added to the beads and vortexed immediately for 1 min. The reaction was then quenched with 125 mM glycine (pH 8.0) and incubated on ice for 5 min. Beads were then washed x5 with IP lysis buffer (20 mM Tris, 150 mM NaCl, 1% Triton X-100, pH 7.5) and the blocked for 1 hr at RT with 1% BSA in IP lysis buffer. Beads were used immediately or stored overnight at 4 °C. The next day, neutrophils (1x107) were seeded in Petri dishes with 10 ml of medium, left unstimulated or stimulated with 100 nM PMA for 3 hr at 37 °C. After stimulation, cells/NETs were washed (x3) very gently with equal volume of PBS (plus 1 mM AEBSF). After the last wash NETs were digested in 1 ml of Benzonase buffer (20 mM Tris-HCl, pH 7.6, 2 mM MgCl2, 1 mM CaCl2) with 250 U benzonase (Sigma E1014) and 1 mM AEBSF, 20 µM NEi and CGi for 30 min at 37 °C, with intermittent rocking to distribute the buffer. The reaction was stopped with the addition of 4 mM EDTA and 4 mM EGTA. Cells were further lysed with the addition of 300 µl of 5 X IP lysis buffer (plus 5 X protease inhibitors). Dishes were incubated on ice for 10 min. Cells and NETs were then collected by scraping into an Eppendorf tube. For multiple dishes at the same time point and stimulation, the samples were pooled and divided for the IP step which was performed immediately. For each IP, lysate from 1x107 cells was used with 15 µl of coupled beads and incubated overnight at 4 °C with gentle rotation. The next day beads were washed 5 times with IP lysis buffer plus inhibitors. After the final wash beads were resuspended in 30 µl 1 X SDS-PAGE sample loading buffer. Samples were analysed by SDS-PAGE and stained or immunoblotted

as indicated. Coomassie Instant blue stain or Thermofisher silver staining kit were used according to the manufacturer's instructions.

## Peptide competition assay

Neutrophils were seeded on coverslips in a 24 well dish (1x105 cells/well) and stimulated with PMA for 3 hr and fixed with 2% PFA. Prior to staining, 3D9 (2 µg/ml = ~12.9 nM) was preincubated overnight with rotation at 4 °C in PBS with 0.01–200 X fold molar excess of branched REIRR peptide. TGGVK branched peptide was used as negative control. The next day competed antibody-peptide solutions were diluted 1:1 with blocking buffer and the immunostaining procedure was followed as before. Peptide competition was assessed visually by fluorescent microscopy.

## Epitope mapping

Antibody binding to human H3 was assessed by a combination of linear, conformational and amino acid replacement analysis epitope mapping with peptide synthesis and ELISA assays performed by Pepscan Presto B.V. (Leiden, The Netherlands). To reconstruct epitopes of the target molecule (H3 residues 30–70 - PATGGVKKPHRYRPGTVALREIRRYQKSTELLIRKLPFQRL) a library of peptide-based peptide mimics (*Figure 4—figure supplement 1—source data 1*) was synthesized using Fmoc-based solid-phase peptide synthesis. An amino functionalized polypropylene support was obtained by grafting with a proprietary hydrophilic polymer formulation, followed by reaction with t-butyloxycarbonyl-hexamethylenediamine (BocHMDA) using dicyclohexylcarbodiimide (DCC) with N-hydroxybenzotriazole (HOBt) and subsequent cleavage of the Boc-groups using trifluoroacetic acid (TFA). Standard Fmoc-peptide synthesis was used to synthesize peptides on the amino-functionalized solid support by custom modified JANUS liquid handling stations (Perkin Elmer). Synthesis of structural mimics (conformational analysis) was done using Pepscan's proprietary Chemically Linked Peptides on Scaffolds (CLIPS) technology. The binding of antibody to each of the synthesized peptides was tested in a Pepscan-based ELISA. The peptide arrays were incubated with primary antibody (3D9 or isotype control) at 0.02 µg/ml for linear and conformational analysis and 0.5 µg/ml for amino acid replacement analysis (overnight at 4 °C). After washing, the peptide arrays were incubated with a 1/1000 dilution of rabbit anti-mouse IgG(H+L) HRP conjugate (Southern Biotech, Birmingham, AL, USA) for one hour at 25 °C. After washing, the peroxidase substrate 2,2'-azino-di-3- ethylbenzthiazoline sulfonate (ABTS) and 20 µl/ml of 3% H2O2 were added. After one hour, the colour development was quantified using a charge coupled device (CCD) - camera and an image processing system. The values obtained from the CCD camera ranged from 0 to 3000 mAU, similar to a standard 96-well plate ELISA-reader. The results were quantified and stored in the Peplab database. To verify the quality of the synthesized peptides, a separate set of positive and negative control peptides was synthesized in parallel. These were screened with commercial antibodies 3C9 and 57 (*Posthumus et al., 1990*).

To evaluate the binding and determine the core epitope, screenings were optimized for concentration and blocking conditions for each sample until clear signals over background values were observed. Peaks were defined as at least in the third quartile for the specific peptide mimic values. In addition, a binding event was only noted if multiple overlapping peaks were present within the binding region. To determine a core epitope, peptides within 40% of the main peak intensity were evaluated.

## Quantification of staining characteristics by Image J and R

In order to assess staining characteristics of antibodies during NET formation we developed a bundle of Image J and R scripts. These scripts allow for an automated workflow starting from 2-channel microscopic images of an experimental series (DNA stain, antibody stain), to a graphical representation and classification of individual cells and eventually to mapping these classifications back to the original images as a graphical overlay. In the first step, nuclei are segmented based on intensity thresholding (either programmatic or manual), including options for lower and upper size selection limits. The same threshold is applied to the entire experimental series and the upper size limit is used to exclude fused structures that cannot be assigned individual cells. A quality score is assigned to every image based on the fraction of the total DNA stained area that can be assigned to individual cells (or NETs). This score along with all other parameters of the analysis is exported as report file and can be used to automatically exclude images from the analysis. In addition, this part of the script generates a result file that includes the area, circumference (as x,y coordinates) and cumulative intensities for each

channel for every detected nucleus along with information such as time point or stimulus that can be assigned programmatically. In order to analyse these data sets, we implemented a series of functions in R. These functions include import of Image J result files, classification of cells based on nuclear area and staining intensity, various plot and data export functions, as well mapping functions that allow to display the classification of nuclei as color coded circumferences overlaid on the original images. The scripts are available for download at https://github.com/tulduro/NETalyser; copy archived at swh:1:rev:3de3df622e7c4b8622273355b4a6176aa0157db3, *Tilley, 2021*.

## Quantification of NETs by Image J

NETs were quantified by the semi-automatic method described by *Brinkmann et al., 2012* and via a second modified method that allowed automatic quantification. All microscopy image datasets were processed by both methods to allow comparison. Hoechst was used to stain total DNA and NETs were additionally stained with anti-chromatin (PL2.3) or 3D9 antibodies (1 µg/ml) and Alexa-568 coupled secondary antibody according to the previous section. Images were acquired with a Leica DMR upright fluorescence microscope equipped with a Jenoptic B/W digital microscope camera and 10 x (numerical aperture 0.30) or 20 x (numerical aperture 0.50) objective lens. For each experiment, the same exposure settings were used for all samples and a minimum of 3 random fields of view (FOV) were collected. Images were analysed using ImageJ/FIJI software. As described by *Brinkmann et al., 2012*, each channel was imported as an image sequence and converted into a stack. To count total cells/NETs per FOV, the Hoechst channel stack was imported and segmented using the automatic thresholding function (Bernsen method) with radius 15 and parameter 1 set to 35 to produce a black and white thresholded image. Particle analysis was then performed to count all objects the size of a cell nucleus or bigger (10 x objective: particle size 25-infinity; 20 x objective: particle size 100-infinity) and to exclude background staining artefacts. Total NETs were then counted using the anti-chromatin (PL2.3) or 3D9 channels accordingly. For the method published in 2012, anti-chromatin stains were segmented using manually adjusted thresholding so that the less intense staining of resting cell nuclei was excluded. Particle analysis was then performed to count all objects larger than a resting cell nucleus (10 x objective: particle size 75-infinity; 20 x objective: particle size 250-infinity). This was also performed for 3D9. In contrast, in the second analysis, the workflow was modified so that the automatic Bernsen thresholding and segmentation were used for both Hoechst and NET channels, total cells and NETs respectively. For each method percentage NETs were calculated as (NETs/Total cells) x100. Results per FOV were then averaged according to sample. A schematic of the different workflows is presented in *Figure 5—figure supplement 1*.

## Immunofluorescent staining of histological samples from tissue sections

Paraffin sections (2 µm thick) were deparaffinized in two changes of 100% xylene for 5 min each and then rehydrated in two changes of 100% ethanol for 5 min each and followed by 90% and 70% ethanol for 5 min each. Sections were washed with 3 changes of deionized water and incubation in TBS (Tris buffered saline). For antigen retrieval, Target Retrieval Solution (TRS pH9) (Dako S2367) was used to incubate the slides in a steam cooker (Braun) for 20 min. After cooling down to room temperature in antigen retrieval buffer, slides were rinsed 3 x in deionized water and incubated in TBS until further processing. Slides were blocked with blocking buffer (1% BSA, 5% normal donkey serum, 5% cold water fish gelatin and 0.05% Tween20 in TBS, pH7.4) for 30 min. Blocking buffer was removed, and sections were incubated with primary antibodies at appropriate dilution in blocking buffer (containing 0.05% Triton-X100) overnight at RT. Sections were rinsed in TBS and then incubated with secondary antibody at an appropriate concentration (1:100) for 45 min in the dark at RT and then rinsed three times in TBS for 5 min followed by rinsing with deionized water. Slides were incubated with DNA stain Hoechst 33342 (1:5000) for 5 min, rinsed with water before mounting with Mowiol. Primary antibodies for detection of NETs were as follows: mouse anti-cleaved histone 3 clone 3D9 (2 µg/ml); rabbit anti-histone H3 antibody (citrulline R2+R8+R17; ab5103, Abcam); chicken anti-Histone H2B ab134211 Abcam (1:400); sheep anti-ELANE (NE) LS-Bio LSB 4244 Lot 75251. Secondary antibodies were the following: donkey anti-rabbit immunoglobulin G (IgG) heavy and light chain (H&L) Alexa Fluor 488 (Jackson 711-225-152); and donkey anti-mouse IgG H&L Cy3 (Jackson 715-165-151); donkey anti-sheep (IgG) H&L Alexa Fluor 647 (Jackson 713-605-147). An upright widefield microscope (Leica DMR) equipped with a JENOPTIK B/W digital microscope camera or a Leica confocal microscope SP8 were

used for fluorescent imaging. Z-stack images were collected at 63×magnification (Plan Apochromat 63 x/0.75). Where stated colocalisation analysis was performed on confocal images using Volocity 6.5.1 software. Human tonsil (denoted normal but showing neutrophil infiltration) and inflamed kidney paraffin tissue blocks were purchased from AMSbio. Inflamed tissue from a gallbladder and appendix was obtained from archived leftover paraffin embedded diagnostic appendicitis samples and used in an anonymised way after approval through the Charité Ethics Committee (Project EA4/124/19, July 24, 2019). Informed consent from patients for use of biomaterials for research was obtained as part of the institutional treatment contract at Charité.

## Statistics

All experiments were repeated three times unless stated differently in the figure legend. Experimental repeats are biological replicates, where each replicate represents cells isolated from a different donor. All graphs were prepared in GraphPad Prism and are either representative traces or mean ± standard deviation as stated in the figure legend. Graphs for epitope mapping were produced by Pepscan using proprietary software.

## Contributions

DOT- Conceptualization, methodology, investigation, data curation, figure preparation, writing original draft, review and editing; UZA - 2-DE, blotting and sample preparation for protein sequencing; UA -preparation of human pathology tissue samples and microscopy of tissue samples, image curation; MS - MALDI mass spectrometry; SF - provision and description of human pathology tissues and advice on histology figure presentation; PRJ - proteomic and 2-DE analysis; VB - colocalisation analysis, image curation, reviewing and critical feedback on manuscript; AH - conceptualization, data curation for R analysis, R scripting; AZ- conceptualization, reviewing and editing of manuscript; AZ and AH provided critical feedback to DOT and helped shape the research, analysis and manuscript.

## Acknowledgements

The authors would like to give special thanks to Borko Amulic and Gerben Marsman for their constructive feedback at the early stages of manuscript preparation. This work was supported by the Max Planck Society.

---

## Additional information

### Competing interests

Dorothea Ogmore Tilley, Alf Herzig, Arturo Zychlinsky: has made a patent application for this antibody hybridoma cell line and sequence and its use in the detection of NETs outside of research purposes. No. EP 21 159 757.0. The other authors declare that no competing interests exist.

### Funding

| Funder | Grant reference number | Author |
|---|---|---|
| Max Planck Institute for Infection Biology | | Arturo Zychlinsky |

The funders had no role in study design, data collection and interpretation, or the decision to submit the work for publication.

### Author contributions

Dorothea Ogmore Tilley, Conceptualization, Data curation, Formal analysis, Investigation, Methodology, Project administration, Visualization, Writing – original draft, Writing – review and editing; Ulrike Abuabed, Data curation, Investigation, Preparation for and Mass spectrometry of protein samples and interpretation of the results, Immunofluorescence staining and visualisation and interpretation of tissue sections; Ursula Zimny Arndt, Data curation, Investigation, Methodology, 2D-electrophoresis of purified histone fractions and preparation of samples for Edman degradation sequencing; Monika Schmid,

Data curation, Investigation, Writing – review and editing, Preparation for and Mass spectrometry of protein samples and interpretation of the results; Stefan Florian, Resources, Formal analysis, Writing – review and editing, provision of patient tissue samples for immunofuorescent microscopy. Interpretation of tissue section staining; Peter R Jungblut, Data curation, Formal analysis, Project administration, Writing – review and editing, 2D electrophoresis and interpretation of results; Volker Brinkmann, Data curation, Methodology, Writing – review and editing, Microscopy colocalisation analysis; assistance in figure preparation; Alf Herzig, Conceptualization, Investigation, Methodology, Software, Supervision, R scripting for analysis of staining characteristics; data curation for analysis of staining characteristics; Visualisation of analysis of staining characteristics; Provided constructive feedback and mentorship and helped shape the research; Arturo Zychlinsky, Conceptualization, Resources, Supervision, Writing – review and editing, Provided resources

## Author ORCIDs
Dorothea Ogmore Tilley ![ORCID] http://orcid.org/0000-0003-3254-6991
Volker Brinkmann ![ORCID] http://orcid.org/0000-0003-4735-3682
Alf Herzig ![ORCID] http://orcid.org/0000-0003-4246-4666
Arturo Zychlinsky ![ORCID] http://orcid.org/0000-0001-6018-193X

## Ethics
Samples were collected from healthy donors who had provided informed consent according to the Declaration of Helsinki. Ethical approval was provided by the ethics committee of Charité-Universitätsmedizin Berlin and blood was donated anonymously at Charité Hospital Berlin. For histological tissue samples, tissue was obtained from historical archives and used in an anonymised way after approval through the Charité Ethics Committee (Project EA4/124/19, July 24, 2019). Informed consent from patients for use of biomaterials for research was obtained as part of the institutional treatment contract at Charité.

## Decision letter and Author response
Decision letter https://doi.org/10.7554/eLife.68283.sa1
Author response https://doi.org/10.7554/eLife.68283.sa2

---

# Additional files

## Supplementary files
• Transparent reporting form

## Data availability
Data generated or analysed during this study are included in the manuscript. Source data files have been provided.

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
