## [Editor Report]

This study presents and characterizes a new antibody to detect human neutrophil extracellular traps (NETs). The authors identify a NET-specific histone H3 cleavage and make use of this event to develop a cleavage site-specific antibody. This new tool should be useful to investigators interested in detecting and quantifying human NETs.

---

## [Decision Letter]

**Decision letter after peer review:**

Thank you for submitting your article "Histone H3 clipping is a novel signature of Human Neutrophil Extracellular Traps" for consideration by *eLife*. Your article has been reviewed by 3 peer reviewers, and the evaluation has been overseen by a Reviewing Editor and Carla Rothlin as the Senior Editor. The following individual involved in review of your submission has agreed to reveal their identity: Felipe Andrade (Reviewer #2).

Essential revisions:

Please note the following specific points that need to be clarified.

The authors should demonstrate that 3D9 detects ROS-independent NETs induced by at least two well-described stimuli associated with human diseases (of note, data on platelets as drivers of ROS-independent NETs is puzzling and will need confirmation by the authors if this stimulus is used in these assays).

There are several very well-defined autoimmune stimuli that are potent inducers of NETs, which can be easily reproduced in vitro. In particular, the authors should validate 3D9 using NETs induced by anti-RNP immune complexes (Sci Transl Med. 2011, 3:73ra20) and by rheumatoid factor (Sci Immunol 2017, 2:eaag3358).

Can the authors show if the 3D9 antibody detects so-called "vital NETs"?

Can the authors clarify if the structures induced by *C. albicans* are NETs at all or rather remnants of other forms of neutrophil death?

Can you the authors show how 3D9 perform in serum samples?

Figure 7 should also include the analysis of H3cit. It is unclear why PL2.3 was only included in this figure to compare with 3D9.

Figure 8 and Figure 8—figure supplement 1 should also include the study of pyroptosis. In addition, analysis of H3cit staining should also be included in Figure 8—figure supplement 1. Defining the specificity of 3D9, PL2.3 and H3cit is important to better understand their patterns of staining in tissues.

To discard that citrullination of H3 blocks its cleavage and detection by 3D9, one possibility could be to determine whether 3D9 detection of NETs induced with calcium ionophores changes in the presence of PAD inhibitors. If H3cit is resistant to cleavage during NETosis, PAD inhibition should increase 3D9 detection particularly at late time points (e.g. 180 mins) in which the H3 fragment is no longer detected by immunoblotting (*eLife*. 2017, 6:e24437, Figure 7).

To compare the specificity of 3D9, PL2.3 and H3cit in tissues, the study should include the analysis of markers of other forms of neutrophil death. I found very unlikely that apoptotic neutrophils are excluded from sites of inflammation. in vitro studies are useful and elegant, but the real validation of anti-NET antibodies compared with other forms of neutrophil death should be done in tissues.

Please include the diagnosis of the disease from where the inflamed tissues were collected.

Additional recommendations:

Lines 215-217 are not clear. Please rephrase.

The authors should indicate more details for the microscopy: What kind of objective was used? Please indicate each time numerical aperture and type of objective e.g. Plan Apochromat.

The Image J procedure should be described more in detail. I am suggesting to add a complete script as Supplemental Material.

The script also need to answer the following questions:

What kind of samples were used for threshold setting? Did the authors use specific positive and/or negative NET control images?

How was the diamater of a resting nucleus defined? Is the value indicated as mean value or maximum value of resting nuclei size?

What happens if NETs derived from different nuclei have contact with each other? I assume Image J is then counting only one event. Is that influencing the result at later time points?

This is for example seen in *Candida albicans* or Heme +TNF-mediated NET formation (Figure 6). Is Image J able to quantify those images properly?

[Editors’ note: further revisions were suggested prior to acceptance, as described below.]

Thank you for resubmitting your work entitled "Histone H3 clipping is a novel signature of Human Neutrophil Extracellular Traps" for further consideration by *eLife*. Your revised article has been evaluated by Carla Rothlin (Senior Editor) and a Reviewing Editor.

The manuscript has been improved but there are some remaining issues that need to be addressed.

To address these remaining issues we suggest that you include a section on "limitations of the study" after the discussion that fully addresses the concerns described below.

*Reviewer #1 (Recommendations for the authors):*

In the revised version, the authors have sufficiently reacted to some, although not all, of the essential revisions that had been required by the reviewers. Many of the revisions constitute rephrasing and additions to the discussion rather than having performed additional experiments. I am a bit underwhelmed by the authors´ unwillingness to thoroughly compare tissue staining patterns of 3D9 with those of other antibodies commonly used for detecting NETs in tissues. I do not agree with the authors that this is outside the scope of the manuscript, because cross-comparison would be very important to evaluate the usefulness and added benefit of this new antibody as a tool for detecting NETs.

I would also have expected to get a more thorough evaluation of the applicability of 3D9 in serum. The information that 3D9 cross-reacts with plasma proteins is valuable, but it does not exclude a potential use of 3D9 for detecting NETs in patients´ sera.

As it stands the manuscript is still a valuable contribution to the field but important questions remain open and need to be addressed in future studies.

*Reviewer #2 (Recommendations for the authors):*

The authors addressed properly to the majority of the comments. However, there are still some points that require better clarification in the manuscript. There is no doubt about the quality of the work, particularly coming from the Zychlinsky lab. Certainly, while the 3D9 antibody still requires more characterization, it is going to be a powerful tool for the analysis of NETs. The problem is not the 3D9 antibody, but the other NET-"specific" markers used for comparison. Although citrullination of histone H3 (H3cit) is considered "a reasonably specific marker for NETs in disease models" (as recently stated by a panel of experts, J Exp Med. 2022, 219:e20220011), unfortunately, this statement is not true. Thus, the clear lack of co-staining of 3D9 and H3cit in tissues can be explained because H3cit is not detecting NETs. The proposal that H3cit and 3D9 detect different time points during NETosis is interesting, but it is not demonstrated. The new time course experiment (figure 11—figure supplement 2) comparing H3cit and 3D9 used PMA to induce NETs, which is a poor inducer of citrullination *eLife*. 2017 Jun 2;6:e24437. It is unclear why this stimulus was used to address this point. No more experiments are required. But, to be fair in the discussion, the authors should examine/discuss the possibility that the staining of H3cit in tissues is not detecting NETs. Reviewer #1 may be correct that 3D9 "…would potentially render a large part of the literature that has used citH3 staining for the detection of "NETs" useless.".

The abnormal and abundant production of NETs as the driver of autoimmunity is currently the hallmark in autoimmune rheumatic diseases, and it is well accepted that the mechanism of NETs induction is mediated by immune complexes. The finding that the Zychlinsky lab was unable to produce NETs using anti-RNP complexes is intriguing, not a criticism, because if there is a place that knows how to study NETs, it is the Zychlinsky lab. Although the study of NETs induced by anti-RNP complexes is out of the scope of this manuscript, this experiment was performed because it was important to validate 3D9. Indeed, compared to PMA or nigericin, anti-RNP-induced NETs is truly critical for human pathology. Therefore, whether the result was positive or negative, it is similarly important and significant. All NETs-inducing mechanisms tested must be mentioned in the text. Unfortunately, 3D9 was unable to be challenged with anti-RNP-induced NETs because this mechanism was not reproducible, and therefore, will require confirmation by others. This caveat must be mentioned somewhere in the text and at least a sentence in the discussion.

---

## [Author Response]

Essential revisions:Please note the following specific points that need to be clarified.The authors should demonstrate that 3D9 detects ROS-independent NETs induced by at least two well-described stimuli associated with human diseases (of note, data on platelets as drivers of ROS-independent NETs is puzzling and will need confirmation by the authors if this stimulus is used in these assays).

We thank the reviewer for the recommendation to look at additional stimuli. We apologize in advance as we have noted a mistake in the manuscript designating the disease relevant stimulus heme as NOX dependent. This has been corrected (line 302) to “*both NOX dependent (PMA) and NOX independent (heme;nigericin) stimuli result in NETs*”. Heme induces NETs in CGD patients lacking a functioning NADPH oxidase complex but intracellular ROS scavengers inhibit heme induced NETs (Knackstedt et al., 2019 DOI: 10.1126/sciimmunol.aaw0336) -thus this stimulus is ROS dependent while being independent of the NOX induced ROS burst, with the ROS likely being induced by a chemical reaction with heme itself. For this confusion over the designation of ROS dependency, we categorise our stimuli based on their NOX (in)dependency.

In this manuscript we tested two NOX dependent and two NOX independent stimuli: these are the mitogen PMA and infections with *Candida albicans* (NOX dependent), and heme plus TNF and nigericin (NOX independent). *C. albicans* is a medically relevant pathogen. Heme, a product abundant in malaria, and TNF are a model for cytokine activation in Plasmodium infections. Nigericin, while not yet used in humans, is used in veterinary medicine and examined as an anti-tumor agent for the treatment of human colon cancer among others (DOI 10.1158/1535-7163.MCT-17-0906) . Both PMA and Nigericin are strong inducers of the two canonical pathways to NET formation – NOX dependent and independent respectively.

In deciding to publish this research as a tools and resources paper in *eLife*, we are committing to making this antibody available to the research community for further exploration of NETs in disease contexts. We believe that 3D9 will be useful in exploring NETs in a range of diverse disease contexts, of which other research groups are best placed to assess its performance.

There are several very well-defined autoimmune stimuli that are potent inducers of NETs, which can be easily reproduced in vitro. In particular, the authors should validate 3D9 using NETs induced by anti-RNP immune complexes (Sci Transl Med. 2011, 3:73ra20) and by rheumatoid factor (Sci Immunol 2017, 2:eaag3358).

We have tried to induce NET formation using anti-RNP antibodies as per Sci Transl Med. 2011, 3:73ra20 but we were not successful.

Can the authors show if the 3D9 antibody detects so-called "vital NETs"?

The phenomenon of “vital NETs” and whether histone H3 cleavage at H3R49 occurs during this process would be a very interesting question to explore but, unfortunately, we are not yet able to investigate this using this antibody. The occurrence of vital NETs is a rarely observed event. Intravital and live cell imaging studies provide the best evidence for this (Yipp et al., 2013, doi: 10.1182/blood-2013-04-457671). However, our antibody selection method was optimised for use with fixed or denatured human neutrophil samples only.

We have addressed this limitation and its consequences for choice of assay in the discussion in the new manuscript.

Line 315-318

“….thus, care should be taken in the design of future assays and selection of sample when detecting cleaved H3, NETs or vital NETs under native and mild detergent conditions. In particular, 3D9 alone is not suitable for direct detection of NETs in complex biological fluids and a sandwich approach or colocalization with a neutrophil granule protein is critical.”

In publishing with *eLife* we commit to making this antibody available to serve the research community who may chose to engineer it in a way that could make it useful for live cell imaging or whole blood samples for research purposes.

Can the authors clarify if the structures induced by C. albicans are NETs at all or rather remnants of other forms of neutrophil death?

For this study we use the histological definition of NETs as extracellular chromatin decorated with neutrophil granule proteins. Thus, we can describe these visible structures as NETs as they stain for DNA and neutrophil elastase. It is possible that they are citrullinated NETs, as observed by Kenny et al., (*eLife*. 2017, 6:e24437, Figure 6). However, we cannot exclude that these are remnants of other forms of neutrophil cell death and this remains a problem faced by the entire field as recently reported by Boeltz et al., 2019 (DOI:10.1038/s41418-018-0261-x).

We have expanded on this observation in the discussion: line 281-284

“Like citrullination, not all NETs contain H3 cleaved at H3R49. With *Candida albicans*, some NET-like structures were not 3D9 positive (Figure 6). These may be remnants of other forms of cell death or they may be citrullinated NETs as has been shown by Kenny et al., (2017).”

Can you the authors show how 3D9 perform in serum samples?

This is an informative experiment for users of this antibody and we thank the reviewers for recommending to include it. In preparation for using the antibody with blood samples, we performed some preliminary experiments to address whether the antibody was suitable for a simple ELISA in the presence of plasma/serum. We have now added Figure 3—figure supplement 4 addressing this. In this experiment**,** we examined the ability of 3D9 to react with healthy donor plasma and serum alone by direct ELISA Figure 3—figure supplement 4 (A) (biological replicate n=1). 3D9 strongly reacted with plasma but not with serum. We also examined detection by 3D9 of western blotted serum and plasma proteins separated by SDS page under reducing and non reducing conditions Figure 3—figure supplement 4 (B) and found that 3D9 also detected proteins, of higher molecular weight than cleaved H3, in plasma but not in serum. We conclude that the direct detection of NETs by 3D9 in plasma containing samples and thus whole blood is not possible due to cross reaction with a plasma protein(s). Based on this we caution the use of 3D9 in serum containing samples alone as it is challenging to ensure all plasma proteins are removed and instead we advocate for the use of sandwich and colocalization approaches.

In the main text in we have included the following lines 152-156

“However, when we performed preliminary experiments to see if the antibody had the potential to work in blood samples, we observed a strong reaction of the antibody with a plasma protein(s), but not with serum-protein(s) as determined by direct ELISA and western blot Figure 3—figure supplement 4. Therefore, 3D9 is not suitable for direct detection of NETs in biological fluids that may contain plasma proteins.”

In the discussion Line 314-319

“Furthermore, a preliminary investigation revealed 3D9 reacts with a plasma protein(s) in ELISA and western blot – albeit of a higher molecular weight – (Figure 3—figure supplement 4A) and thus, care should be taken in the design of future assays and selection of sample when detecting cleaved H3, NETs or vital NETs under native and mild detergent conditions. In particular, 3D9 is not suitable for direct detection of NETs in complex biological fluids and a sandwich approach or colocalization with a neutrophil granule protein is critical.”

And appropriate text has been added to the methods section line 567-575

“The same approach was used to examine 3D9 interactions with immobilised serum and plasma proteins. Plasma was isolated from whole blood collected with S. Monovette sodium citrate tubes (Sarstedt). Whole blood was centrifuged at low speed to minimise cell lysis (150 xg, 20 min with no brake). Prostaglandin E1 (1 µm) was added to inhibit platelet activation and samples were further centrifuged at 650 xg (8 min) to collect cell free plasma and further centrifuged at 2000 xg (10 min) before being aliquoted and stored at -80 °C. Serum was isolated by collection of whole blood in S. Monovette serum tubes (silicate clotting activator) and incubation with gentle rotation for 30 min at RT before centrifugation at 2000 xg at 4°C (10 min) and collection of serum.”

Figure 7 should also include the analysis of H3cit. It is unclear why PL2.3 was only included in this figure to compare with 3D9.

In this figure PL2.3 was used as a control to detect all NETs co-staining with NE. Chromatin stains are regularly used in the literature to facilitate detection of NETs, albeit with colocalization of a neutrophil granule protein as done here. PL2.3 alone is not sufficient to detect NETs (nor is H3cit) and literature which uses a chromatin marker alone to detect NETs is not, in our opinion, evidence of NETs in vitro or in tissues.

We have added a statement to clarify the use of PL2.3 in combination with anti-NE as a control for in vitro experiments throughout the manuscript and this is reiterated in the experimental methods

Line 186-190

“We compared its staining to a control antibody commonly used in combination with neutrophil markers to detect NETs – PL2.3, an anti-chromatin antibody directed against a H2A-H2B-DNA epitope (Losman et al., 1992). PL2.3, in conjunction with staining for neutrophil granule proteins, is used to facilitate detection of NETs by immunofluorescent microscopy (Brinkmann et al., 2012).”

Line 228-231

“Importantly, 3D9 detected only nuclei that appeared decondensed in cells that were positive for NE, a specific neutrophil marker. In contrast, the control chromatin antibody stained both neutrophils in NETosis (co-staining with anti-NE) and the nuclei of other cells (lacking NE).”

Line 443-444 (methods)

“PL2.3 colocalisation with anti-NE was used as a control for NET detection in all in vitro experiments.”

We have expanded further on our rational for not making a direct comparison of H3 cit and 3D9 staining further in this response.

Figure 8 and Figure 8—figure supplement 1 should also include the study of pyroptosis.

We thank the reviewer for this suggestion and agree that pyroptotic cell death would be interesting to examine in neutrophils with respect to distinguishing different forms of neutrophil cell death from NETosis with 3D9. However the variation in neutrophil responses to pyroptotic stimuli would warrant a deeper investigation beyond the scope of this manuscript. Unlike macrophages, human neutrophils are largely resistant to pyroptotic cell death when inflammasome pathways are activated Chen et al., 2014, Chen et al., 2018, Karamakar et al., 2020, Kovacs et al., (Cell Rep. 2020 Jul 28; 32(4): 107967. ). Stimulation or infection of neutrophils with macrophage pyroptotic stimuli, intracellular pathogen or pathogen signals within the cytosol, elicits some of the characteristics of pyroptosis but not the typical cell morphology. In macrophages pyroptosis is preceded by inflammasome (canonical and non-canonical) activation resulting in activation of specific caspases, caspase mediated activation of Gasdermin D and assembly of a gasdermin pore in the plasma membrane resulting in disruption of osmostic regulation and rapid cell death and the concomitant release of cytokines including IL-1b/ IL-18 that are matured through caspase activity. In contrast, in neutrophils, inflammasome activation by infection with intracellular pathogens can produce diverse outcomes in terms of cell death depending on the specific intracellular pathogen or signal. While *Salmonella* triggers release of caspase 1 activated IL-1beta from neutrophils (Chen et al., 2014, https://doi.org/10.1016/j.celrep.2014.06.028) it does not induce pyroptosis. Infection with *Citrobacter rodentium* triggers release of IL-β but goes on to produce NETs in a manner that is inflammasome driven and caspase dependent (Chen et al., 2018 Science Immunology) (https://immunology.sciencemag.org/content/3/26/eaar6676.long). Most recently Kovacs et al., have shown that instead of oligermising at the plasma membrane, Gasdermin D forms pores in azurophilic granules and autophagolysome and that IL-1b release involves autophagy machinery. Thus, given the variability established in neutrophil pyroptotic-like responses and their inherent resistance to undergo typical pyroptotic cell death, it is unclear what value examining pyroptotic stimuli will add in the context of characterising 3D9 staining of NETs. Indeed, in Figure 6 we use Nigericin as a stimulus to induce NETs which are detected by 3D9 and it is a known activator of the inflammasome pathway and pyroptosis in macrophages but produces NETs in neutrophils.

To make it clear that we have not exhaustively looked at all forms of neutrophil cell death we have modified the section title, figure titles and main text body to reflect that we have looked at apoptotic, necroptotic and necrotic stimuli.

line 233-234

“3D9 distinguishes NETosis from apoptotic, necroptotic and necrotic cell death in neutrophils.”

line 273-274

“…and distinguishes netotic neutrophils from neutrophils that die via apoptosis, necrosis and necroptosis, in vitro.”

In addition, analysis of H3cit staining should also be included in Figure 8—figure supplement 1. Defining the specificity of 3D9, PL2.3 and H3cit is important to better understand their patterns of staining in tissues.

This response also applies to revision 5.

We thank the reviewers for their critical understanding of the field and their suggestion to add a comparison of staining with H3cit to Figures 7 and 8. Including comparison stains with PL2.3 will always be useful in ensuring we are examining all areas of putative NETs, and staining for histone citrullination and/or histone H3R49 cleavage will add specificity as they detect processes that take place during NET formation. However, from the outset of this manuscript we did not set out to qualitatively compare H3cit as a surrogate marker of NETs to our new antibody. It is possible that other mechanisms of cell death may involve citrullination but our aim in this study was to characterise 3D9 staining behaviour in varied contexts – not H3cit behaviour, which is an ongoing area of debate in the research community - Boeltz et al., 2019 (DOI:10.1038/s41418-018-0261-x).

Through our investigation into the precise site of the histone cleavage we determined that, by nature of the epitopes of 3D9 and the most commonly used abcam H3cit R2, R8, R17 antibody, they cannot stain the same individual histone. Thus, in theory, examining the staining patterns of 3D9 and H3cit, side by side, as we have done for PL2.3, will not provide insight as to which NETs or other modes of cell death involve citrullination – only whether H3cit is present or absent.

The finding, that individual neutrophils or NETs showed such ‘either or’ – H3cit or cleaved H3 – characteristics as seen in the tissue sections was surprising and intriguing and this has led us to propose that different types of NETs, specifically those that are more proteolytically processed at the H3 N-terminal, or as a reviewer has suggested, different degrees of citrullination, may be being distinguished by 3D9. However, this will need further investigation that goes beyond the scope of this manuscript and will need to examine other citrullination sites, e.g. H4cit or even pan citrullination to determine if 3D9 distinguishes NETs with more or less citrullination.

What we have developed in this manuscript is an additional antibody, another tool in the arsenal of NET researchers, to look at NETs (that may or may not also have been citrullinated at H3 R2, 8 or 17 or at a different histone during the process of NET formation). It is another tool for detection. We examine H3cit, 3D9 and H2B co-staining in our final tissue section figures and in doing so demonstrate the diversity of NETs in tissues and highlight how it is important to follow the histological definition of NETs and not rely on a single marker. Individually each antibody has failings but researchers who make use of multiple methods to assess NETs can be confident of their assessment of NETs in their studies.

To discard that citrullination of H3 blocks its cleavage and detection by 3D9, one possibility could be to determine whether 3D9 detection of NETs induced with calcium ionophores changes in the presence of PAD inhibitors. If H3cit is resistant to cleavage during NETosis, PAD inhibition should increase 3D9 detection particularly at late time points (e.g. 180 mins) in which the H3 fragment is no longer detected by immunoblotting (eLife. 2017, 6:e24437, Figure 7).

This is an important line of logic to follow for the field’s mechanistic understanding of how citrullination may affect H3 cleavage dynamics however we think this investigation is beyond the scope of this manuscript. However, we have some interesting unpublished data we can share with the reviewer.

Indeed, it is quite possible that citrullination may affect the kinetics of histone cleavage, enhancing or inhibiting it. While we have not used the calcium ionophore in this study, we have used another ionophore for potassium, nigericin, and it induces maximum H3 citrullination around 2h as we can see in a preliminary experiment (Author response image 1). Interestingly, inhibition of PAD 2/4 by BB CL^-^Amidine completely inhibited citrullination but did not delay histone H3 cleavage. This is in line with Kenny et al., *eLife*. 2017, 6:e24437, Figure 6, who showed that PAD inhibition had no effect on nigericin induced NETs. Together, in the case of a K^+^ ionophore, and inflammasome activator, citrullination of H3 does not block H3 cleavage and this cleavage is detectable by 3D9 in immunofluorescence (Figure 6). Citrullination might however, regulate the extent of histone proteolysis with almost complete cleavage of the full length H3 in the presence of inhibitors of citrullination at 2 and 3 hrs versus in the absence of inhibitors.

**Author response image 1. sa2fig1:** 

To compare the specificity of 3D9, PL2.3 and H3cit in tissues, the study should include the analysis of markers of other forms of neutrophil death. I found very unlikely that apoptotic neutrophils are excluded from sites of inflammation. in vitro studies are useful and elegant, but the real validation of anti-NET antibodies compared with other forms of neutrophil death should be done in tissues.

We very much agree with the reviewer that apoptotic neutrophils are indeed likely to be at the site of inflammation and we have added a statement to our Results section to reflect the varied pathways a neutrophil may take at a site of inflammation.

line 246-247.

“Neutrophils are recruited to sites of inflammation and depending on the context or the surrounding stimuli, they may undergo varied forms of cell death.”

We attempted to re-examine our tissue sections and stained for cleaved caspase 3 as a marker of apoptosis however as these tissues had already had 4 or 5 stains and antibodies applied the results were not interpretable. We have added an examination of 3D9 and cleaved caspase 3 as a marker of apoptosis in purified neutrophils (Figure 8—figure supplement 2) showing that 3D9 and cleaved caspase are mutually exclusive on a cellular level. We are aware that in vitro studies do not represent the best method to validate if 3D9 can distinguish cell death in tissues but our in vitro evidence indicates this is likely.

The following was added to the main text body to refer to the new Figure 8—figure supplement 2.

line 241-242

“Further staining of the apoptotic marker, cleaved caspase 3, and 3D9 (Figure 8—figure supplement 2), showed that apoptotic cells did not display cleaved H3R49.”

Please include the diagnosis of the disease from where the inflamed tissues were collected.

A diagnosis of appendicitis has now been included for the appendix and gall bladder samples in the figure titles and legends and further detail is provided in the methods. Some samples were from commercial sources and the provided description of the tissues did not detail the condition the donor had.

line 1086

“Figure 9….(A) human tonsil, denoted ‘normal’ by commercial provider but showing infiltration of neutrophils demonstrating an inflammatory event. (B and C) human kidney, denoted ‘inflamed’ by commercial provider.”

line 1089

“Figure 10. Comparison of Clipped H3, H3cit and H2B staining in the gallbladder from an appendicitis patient.”

Line 1094

“Figure 11. Comparison of Clipped H3, H3cit and H2B staining in the appendix of an appendicitis patient”

Methods section

Line 722-725

“Human tonsil (denoted normal but showing neutrophil infiltration) and inflamed kidney paraffin tissue blocks were purchased from AMSbio. Inflamed tissue from a gallbladder and appendix was obtained from archived leftover paraffin embedded diagnostic appendicitis samples.”

Additional recommendations:Lines 215-217 are not clear. Please rephrase.

Original: “In such assays acetylation is often used to neutralise the contribution of the amino terminal charge. However, to mimic any potential charge created at the newly revealed N-terminus, R49, we also included arrays of unmodified peptides.”

Line 164-169

Revised: “In epitope mapping, acetylation is commonly used during peptide synthesis to neutralize the positive charge of the terminal amine groups, making the peptide more closely resemble its native conformation, as part of a larger protein. However, as we were interested in a binding site formed as a consequence of proteolysis, we included both acetylated and non-acetylated arrays of peptides to allow for potential changes in the charge of the terminal amino acid that may contribute to antibody binding.”

The authors should indicate more details for the microscopy: What kind of objective was used? Please indicate each time numerical aperture and type of objective e.g. Plan Apochromat.

Further details have now been provided in each figure legend.

In general, confocal microscopy 63x Plan Apochromat, 1.30 NA; 20x Plan Apochromat, 0.75 NA. For standard fluorescent microscopy 10x Fluotar 0.30 NA, 20x Fluotar, 0.50 NA

The Image J procedure should be described more in detail. I am suggesting to add a complete script as Supplemental Material.

The Image J quantification procedure (Figure 5C) has been published by Brinkmann in 2012 as a stand alone methods paper.

The new analysis method for describing the characteristics of staining (Figure 5B) is a script bundle written in Image J macro language and R. Both NETalyser scripts are available at https://github.com/tulduro/NETalyser

This link is included in the method section – line 669

The script also need to answer the following questions:What kind of samples were used for threshold setting? Did the authors use specific positive and/or negative NET control images?

For the already published image J procedure (Brinkmann 2012), the built-in automatic and manual thresholding functions in image J were used on resting neutrophils and PMA induced NETs respectively (Figure 5B).

For the new analysis method describing the characteristics of staining (NETalyser, Figure 5C) we used sets of samples each consisting of resting neutrophils and a time course after PMA induction. Within each set, segmentation of images into individual structures (nuclei or NETs) was based on the DNA staining by applying intensity and size thresholds. In the script these parameters were determined interactively with one image and then applied to the entire set. The intensity threshold was set manually. The size thresholds were set manually as lower limit (excluding debris), and upper limit (excluding NET structures fused from multiple cells). Images in which less then 70% of the total area stained by the DNA dye was allocated to individual structures (nuclei or single NETs) were automatically excluded from the analysis (see also below). The Image J script exports data on the size, circumference and signal intensity/area for each channel and each individual structure. For the data shown in Figure 5C we normalized staining intensity within each set and plotted these intensities against the size of the structures.

How was the diamater of a resting nucleus defined? Is the value indicated as mean value or maximum value of resting nuclei size?

For the already published image J procedure (Brinkmann 2012), the particle analysis parameters – pixel numbers relating to neutrophil nucleus/NET size, were deduced iteratively for resting neutrophils and NETs, for the microscope used in the experiments. This is described in the previous publication.

What happens if NETs derived from different nuclei have contact with each other? I assume Image J is then counting only one event. Is that influencing the result at later time points?This is for example seen in Candida albicans or Heme +TNF-mediated NET formation (Figure 6). Is Image J able to quantify those images properly?

In the basic Image J methodology (Brinkmann 2012), time points after 4h are often not analysable for the reason the reviewer points out. For some stimuli this point may be reached earlier or later and it is up to the user to assess if Image J is producing an accurate quantification. For both heme and candida as mentioned, the appropriate cell density/concentration/MOI was determined for each experimental set up.

For the analysis of staining characteristics using Image j and R, an upper limit on NET size was applied as part of the script. Images that showed a high degree of fused structures (>30% of all DNA stained area) were altogether excluded from the analysis. For the plots shown in Figure 5C structures exceeding the upper size limit were removed. To assess whether the size limit excluded fused structures accurately we used a feature of our script that allows to automatically highlight these structures in the original images. This is outlined in the methods section.

[Editors’ note: further revisions were suggested prior to acceptance, as described below.]

Reviewer #1 (Recommendations for the authors):In the revised version, the authors have sufficiently reacted to some, although not all, of the essential revisions that had been required by the reviewers. Many of the revisions constitute rephrasing and additions to the discussion rather than having performed additional experiments. I am a bit underwhelmed by the authors´ unwillingness to thoroughly compare tissue staining patterns of 3D9 with those of other antibodies commonly used for detecting NETs in tissues. I do not agree with the authors that this is outside the scope of the manuscript, because cross-comparison would be very important to evaluate the usefulness and added benefit of this new antibody as a tool for detecting NETs.I would also have expected to get a more thorough evaluation of the applicability of 3D9 in serum. The information that 3D9 cross-reacts with plasma proteins is valuable, but it does not exclude a potential use of 3D9 for detecting NETs in patients´ sera.As it stands the manuscript is still a valuable contribution to the field but important questions remain open and need to be addressed in future studies.

We appreciate that this manuscript does not answer many of our burning questions about the detail of the NET landscape but we hope these will be answered by future studies using 3D9 in the near future. We agree the progression of the work will benefit from a deeper investigation and comparison to existing NET staining methods, ideally including but also beyond H3cit e.g. H4cit, and potentially other citrullinated proteins to help elucidate the difference between NETs formed with or without activation of PADs. This will be better served by a subsequent study. We have now highlighted the weaknesses the reviewers have raised and added the importance of the direction the future studies should take, as suggested by the reviewers’ critique, to both the discussion and a new ‘limitations of the study’ section.

With regards the potential use of 3D9 with patient sera, we agree the study does not exclude a use for 3D9 with patient sera and we have modified the results, discussion and new limitations section to emphasise that, while 3D9 may be not suitable for direct detection is plasma containing samples, future studies will be needed to demonstrate if its useful in patient sera which will be supported by colocalization or sandwich approaches.

Results: Line 155

“Therefore, 3D9 may not be suitable for direct detection of NETs in biological fluids that may contain plasma proteins”

Line 324 (Discussion)

“In particular, 3D9 alone may not be suitable for direct detection of NETs in complex biological fluids and a sandwich approach or colocalization with a neutrophil granule protein is likely to be critical.”

Line 349 (limitations)

“Limitations of the study

While this study has attempted to characterise 3D9 as tool to detect human NETs, it is limited by the range of NET inducing stimuli tested. In particular, we were unable to produce NETs in response to immune complexes and it will be important to examine the usefulness of 3D9 in future studies into aberrant NET production in the pathophysiology of autoimmune disease among others. Critically, its usefulness with patient serum samples will need to be demonstrated, bearing in mind the cross reaction with a plasma protein. While here we compared 3D9 staining to another NET detection method – NE and chromatin antibodies – in vitro, and included another NET surrogate marker – H3cit – in tissues, a more in depth cross-comparison between 3D9, H3cit and other NET staining methods will be needed to fully understand the NET landscape and usefulness of 3D9 for tissue samples going forward. Giving consideration to the caveats that have arisen for the use of H3cit, and that H3cit may not be detecting NETosis in all instances, it will be important to identify the protease responsible for cleavage at H3R49 and, in turn, examine chromatin and cell death responses in cells expressing this protease to help fully establish the specificity of H3R49 clipping in NETosis.”

Reviewer #2 (Recommendations for the authors):The authors addressed properly to the majority of the comments. However, there are still some points that require better clarification in the manuscript. There is no doubt about the quality of the work, particularly coming from the Zychlinsky lab. Certainly, while the 3D9 antibody still requires more characterization, it is going to be a powerful tool for the analysis of NETs. The problem is not the 3D9 antibody, but the other NET-"specific" markers used for comparison. Although citrullination of histone H3 (H3cit) is considered "a reasonably specific marker for NETs in disease models" (as recently stated by a panel of experts, J Exp Med. 2022, 219:e20220011), unfortunately, this statement is not true. Thus, the clear lack of co-staining of 3D9 and H3cit in tissues can be explained because H3cit is not detecting NETs. The proposal that H3cit and 3D9 detect different time points during NETosis is interesting, but it is not demonstrated. The new time course experiment (figure 11—figure supplement 2) comparing H3cit and 3D9 used PMA to induce NETs, which is a poor inducer of citrullination (eLife. 2017 Jun 2;6:e24437). It is unclear why this stimulus was used to address this point. No more experiments are required. But, to be fair in the discussion, the authors should examine/discuss the possibility that the staining of H3cit in tissues is not detecting NETs. Reviewer #1 may be correct that 3D9 "…would potentially render a large part of the literature that has used citH3 staining for the detection of "NETs" useless.".

We thank the reviewer for this critique and look forward to the future work in unravelling the different staining of NETs, or as it may be, not NETs. We have now included the following statements in the discussion and limitations section to consider the plausible situation that H3cit R2, 8, 17 may not detect NETs and that it is likely we need to re-evaluate some of the previous work done on H3cit and NETs when no other method was used to detect NETs. This is in addition to some of the known controversaries of H3cit we have already raised in the discussion and we now explicitly refer the reader to the in depth discussion of this by Konig and Anrade (2016).

We have stopped short of saying that anti-H3cit does not detect NETs as, minimally, in the tissue sections presented in this study, H3cit R2, R8, R17 colocalises with areas of decondensed chromatin (DNA/H2B), bearing neutrophil elastase, and thus detects NETs – albeit not the NETs detected by 3D9, bearing the same NET components but whose detection could be excluded by restriction of the antigen present. And, while we have attempted to assess in depth that cleavage at H3R49 is unique to NETs and neutrophils, it may occur during other biological processes that we have not yet considered. We have raised this the new section, Limitations of the study.

Line 251 (Results)

“Citrullination of H3 is also used in NET detection, albeit more convincingly when co-stained with a neutrophil granule or cytoplasmic marker”

Line 283 (Discussion)

“The use of H3cit for the detection of NETs is not without controversy, as discussed in depth by Konig and Anrade (2016).”

Line 299 (Discussion)

“These findings also suggest a need to re-evaluate NET studies derived from experiments using H3cit as the only method used to detect NETs.”

Line 355 (limitations)

“While here we compared 3D9 staining to another NET detection method – NE and chromatin antibodies – in vitro, and included another NET surrogate marker – H3cit – in tissues, a more in depth cross-comparison between 3D9, H3cit and other NET staining methods will be needed to fully understand the NET landscape and usefulness of 3D9 for tissue samples going forward. Giving consideration to the caveats that have arisen for the use of H3cit, and that H3cit may not be detecting NETosis in all instances, it will be important to identify the protease responsible for cleavage at H3R49 and, in turn, examine chromatin and cell death responses in cells expressing this protease to help fully establish the specificity of H3R49 clipping in NETosis.”

The abnormal and abundant production of NETs as the driver of autoimmunity is currently the hallmark in autoimmune rheumatic diseases, and it is well accepted that the mechanism of NETs induction is mediated by immune complexes. The finding that the Zychlinsky lab was unable to produce NETs using anti-RNP complexes is intriguing, not a criticism, because if there is a place that knows how to study NETs, it is the Zychlinsky lab. Although the study of NETs induced by anti-RNP complexes is out of the scope of this manuscript, this experiment was performed because it was important to validate 3D9. Indeed, compared to PMA or nigericin, anti-RNP-induced NETs is truly critical for human pathology. Therefore, whether the result was positive or negative, it is similarly important and significant. All NETs-inducing mechanisms tested must be mentioned in the text. Unfortunately, 3D9 was unable to be challenged with anti-RNP-induced NETs because this mechanism was not reproducible, and therefore, will require confirmation by others. This caveat must be mentioned somewhere in the text and at least a sentence in the discussion.

We have now included a statement of the results of our anti-RNP tests in the results and referred to the small range of stimuli tested as a limitation of the study.

Line 223 (results)

“To test 3D9 with immune complex induced NETs, we attempted to induce NETs with RNP/anti-RNP (ribonucleoprotein) complexes, however there was no induction of NETs in healthy neutrophils.”

Line 349 (limitations)

“While this study has attempted to characterise 3D9 as tool to detect human NETs, it is limited by the range of NET inducing stimuli tested. In particular, we were unable to produce NETs in response to immune complexes and it will be important to examine the usefulness of 3D9 in future studies into aberrant NET production in the pathophysiology of autoimmune disease among others.”